# Limitations of gene editing assessments in human preimplantation embryos

Dan Liang[1,2,17], Aleksei Mikhalchenko[1,17], Hong Ma[1], Nuria Marti Gutierrez[1], Tailai Chen[3], Yeonmi Lee[4,5], Sang-Wook Park[6,16], Rebecca Tippner-Hedges[1], Amy Koski [1], Hayley Darby[1], Ying Li[1], Crystal Van Dyken [1], Han Zhao[3], Keliang Wu [3], Jingye Zhang[3], Zhenzhen Hou[3], Seongjun So[4], Jongsuk Han[4], Jumi Park[5], Chong-Jai Kim[5], Kai Zong[7], Jianhui Gong[8,9], Yilin Yuan[8,9], Ying Gu [8,9], Yue Shen [8,9], Susan B. Olson[10], Hui Yang [11], David Battaglia[1], Thomas O'Leary[12], Sacha A. Krieg[12], David M. Lee[12], Diana H. Wu[12], P. Barton Duell [13], Sanjiv Kaul [13], Jin-Soo Kim [6,14], Stephen B. Heitner[13], Eunju Kang[4,5], Zi-Jiang Chen [1,3,15], Paula Amato[1,12,18] & Shoukhrat Mitalipov [1,18] ✉

Range of DNA repair in response to double-strand breaks induced in human preimplantation embryos remains uncertain due to the complexity of analyzing single- or few-cell samples. Sequencing of such minute DNA input requires a whole genome amplification that can introduce artifacts, including coverage nonuniformity, amplification biases, and allelic dropouts at the target site. We show here that, on average, 26.6% of preexisting heterozygous loci in control single blastomere samples appear as homozygous after whole genome amplification indicative of allelic dropouts. To overcome these limitations, we validate on-target modifications seen in gene edited human embryos in embryonic stem cells. We show that, in addition to frequent indel mutations, biallelic double-strand breaks can also produce large deletions at the target site. Moreover, some embryonic stem cells show copy-neutral loss of heterozygosity at the cleavage site which is likely caused by interallelic gene conversion. However, the frequency of loss of heterozygosity in embryonic stem cells is lower than in blastomeres, suggesting that allelic dropouts is a common whole genome amplification outcome limiting genotyping accuracy in human preimplantation embryos.

Double-strand breaks (DSBs) induced by gene editing are typically repaired by two major mechanisms: nonhomologous end joining (NHEJ) and homology-directed repair (HDR). Repair by the error-prone NHEJ is dominant and frequently leads to relatively small insertions and/or deletions (indels), resulting in small mutagenic alterations at the cleavage site. HDR utilizes endogenous or exogenous homologous sequences as a template to repair cleaved DNA. While critical for gene therapy applications, the frequency of HDR is lower than those of NHEJ[1–4]. Occasionally, DSBs can also lead to larger deletions extending to several thousand base pairs in length but frequency of such outcomes in human embryos remains undetermined[5]. In addition, targeted loci may acquire insertions, inversions, and duplications of DNA segments from other chromosomes or other complex rearrangements.

In human embryos, CRISPR/Cas9 is frequently introduced into a one-celled embryo (zygote) or oocyte during fertilization,

but editing outcomes are evaluated several days later in a 4–8-cell embryo. A common issue observed in most edited mammalian embryos is mosaicism, when an embryo contains sister blastomeres with different on-target alterations[6]. This suggests that the effect of CRISPR/Cas9 and actual repair is likely delayed and occurs after zygotic division, at 2- or 4-cell stage embryos. Mosaicism complicates the analysis of editing outcomes if cells from a multicellular embryo are pooled together for DNA extraction and sequencing. Even deep sequencing of pooled DNA from an embryo may uncover multiple types of on-target edits but fails to reveal frequency of each modification and their allelic distribution. Consequences of NHEJ or HDR with an exogenous template are commonly identified by detection of indel mutations or marker single nucleotide polymorphisms (SNPs). However, loss of heterozygosity (LOH) can be easily overlooked in bulk embryo DNA due to mixture of parental alleles from multiple cells. Another limitation is that genetic analysis of preimplantation embryos requires whole genome amplification (WGA) step to obtain sufficient DNA for sequencing. Most WGA methods, including isothermal multiple displacement amplification suffer from artificial biases. It is estimated that amplification biases and artifacts during single cell WGA can result in up to 30% allelic dropouts (ADO)[7,8]. Consequently, ADO can be interpreted as false-positive deletions, or LOH. In addition, WGA does not preserve chromosomal integrity and yields small fragments typically ranged in size between 0.1 and 10.0 kb[9,10]. Indeed, latest studies observed frequent LOH at the target region following sequence analysis of WGA DNA from human preimplantation embryos[6,11–13]. We suggested that monoallelic DSBs selectively induced on a mutant allele in heterozygous human embryos can be repaired by high-fidelity HDR using intact wild-type homologous allele as template leading to LOH at the target locus[6,14]. A recent mouse study corroborated our conclusions and provided evidence that interhomolog repair occurs in preimplantation embryos via gene conversion without crossover[13]. By contrast, others disputed the possibility of gene conversion and offered an alternative interpretation that LOH observed in human embryos might be caused by large deletions[15,16]. As pointed above, LOH can also result from artificial ADO during WGA leading to amplification of only one allele and giving false positive deletion or gene conversion readouts. Unfortunately, neither sequencing nor computational approaches can compensate for ADO at the locus of interest since the only source of DNA from the original embryonic samples became irrevocably biased.

To resolve these discrepancies and overcome the limitations of analyzing minute embryo DNA, we validated gene editing outcomes in stable embryonic stem cells (ESCs) derived from targeted human embryos. Considering mosaicism in gene edited human embryos, a minimum of 10 clonally propagated ESC lines were established from each blastocyst. This approach allowed a comprehensive analysis of on-target editing outcomes and corresponding frequencies for each modification in ample samples derived from continuously growing cell lines.

## Results

### LOH in human embryos induced by biallelic DSBs
We previously showed that monoallelic DSBs selectively induced on the paternal locus of *MYBPC3* gene carrying pathogenic 4 bp deletion, implicated in hypertrophic cardiomyopathy, resulted in LOH due to loss of the mutant paternal allele[6,14]. To evaluate LOH induced by biallelic DSBs at this locus, here we targeted both wild-type (WT) *MYBPC3* loci (g.14846, NG_007667.1) in homozygous human embryos (Supplementary Fig. 1). Preselected sgRNA along with Cas9 protein and exogenous single-stranded oligodeoxynucleotide (ssODN) were injected into the cytoplasm

of 32 WT oocytes during fertilization with WT sperm (M-phase) and 21 pronuclear stage zygotes 18 h after fertilization (S-phase) (Supplementary Fig. 2a). We introduced 4 synonymous single nucleotide substitutions to the ssODN template to distinguish from the WT allele. Injected oocytes and zygotes were cultured for 3 days, and DNA from individual blastomeres (*N* = 321) of cleaving 4–8 cell stage embryos was processed by WGA to produce sufficient DNA for analyses (Fig. 1a and Supplementary Data 1). To be consistent with previous our and other studies[6,11,12,17] that reported LOH in gene edited human embryos, we used the REPLI-g Single Cell WGA kit from Qiagen utilizing multiple displacement amplification (MDA) method. Following WGA, the target locus was additionally amplified by three long-range PCR primers (1742, 3054, and 8415 bp in size) to detect deletions. PCR products were separated on agarose gels and in addition to the band of expected size, some blastomeres (41/321; 12.8%) showed a secondary band of smaller size, indicative of deletions (Supplementary Fig. 2b). Sequencing analysis of smaller size PCR products suggested deletions ranging in size from one hundred bp to 3.8 kbp (Supplementary Data 1). In most of these blastomeres (31/41, 75.6%), the second allele was presented as a small indel (<100 bp) and thus were designated as *MYBPC3*[Del/Indel]. In addition, one blastomere carried two different large deletions (≥100 bp) at the cleavage site (*MYBPC3*[Del/Del]) while another blastomere showed ssODN in addition to the deletion (*MYBPC3*[Del/HDR]). Interestingly, the remaining 8 blastomeres (8/41, 19.5%) lost the second allele showing the presence of only one large deletion and were designated as *MYBPC3*[homo-Del] (Supplementary Data 1).

Sequencing of remaining individual blastomeres detected only small deletions or insertions less than 100 bp in size that were designated as indel mutations. Again, large portion of these blastomeres (121/321, 37.7%) lost both WT alleles but showed the presence of only one indel mutation (designated as *MYBPC3*[homo-Indel]) (Fig. 1a). Additionally, a few blastomeres (18/321, 5.6%) carried a sequence identical to the ssODN on one or both alleles (*MYBPC3*[WT/HDR], *MYBPC3*[Indel /HDR] or *MYBPC3*[homo-HDR]) indicating HDR with the exogenous template. In addition, 26 blastomeres (8.1%) showed the WT allele only and were deemed as non-targeted *MYBPC3*[homo-WT] (Fig. 1a). We also found that a small portion of blastomeres (8.7%, 28/321) carried one intact WT allele and one indel mutation (classified as *MYBPC3*[WT/Indel]). Remaining blastomeres (27.1%, 87/321) presented two different indel mutations and were designated as compound heterozygous *MYBPC3*[Indel/Indel]. Analysis of sister blastomeres from each embryo revealed that all embryos injected at S-phase (*N* = 21) were mosaic containing more than 2 differently edited blastomeres. In M-phase group, 28 embryos were mosaic and 4 (12.5%, 4/32) contained uniformly edited blastomeres. These results are consistent with our previous observation that injection of sgRNA and Cas9 protein into MII oocytes (M-phase) during fertilization can reduce mosaicism[6] (Supplementary Data 1). Overall, 50.2% blastomeres displayed homozygosity at the *MYBPC3* locus. Remarkably, most homozygosity was de novo and presented as *MYBPC3*[homo-Indel] (37.7%, 121/321), *MYBPC3*[homo-Del] (2.5%, 8/321) or *MYBPC3*[homo-HDR] (1.9%, 6/321) (Fig. 1b).

In an effort to confirm these DNA repair outcomes at different locus, we recruited a sperm donor homozygous for *LDLRAP1* (g.24059 G > A, NG_008932.1) mutation located on chromosome 1 and associated with familial hypercholesterolemia (FH) (Supplementary Fig. 3a). Our initial goal was to target specifically the mutant paternal locus (A) but not maternal WT allele (G) in *LDLRAP1* heterozygous embryos thus inducing monoallelic DSB as opposed to biallelic *MYBPC3* targeting. However, neither of our designed and tested six different sgRNAs was specific to the mutant locus and rather cleaved both maternal and paternal alleles. Therefore, we chose one sgRNA with the highest targeting efficiency for both *LDLRAP1* loci (biallelic)

(Supplementary Fig. 1). CRISPR/Cas9 with ssODN carrying two synonymous single nucleotide substitutions was co-injected with homozygous mutant *LDLRAP1* sperm into the cytoplasm of WT MII oocytes ($N = 19$) (Supplementary Fig. 2c). Injected zygotes along with intact controls ($N = 2$) were cultured for 3 days and individual blastomeres were analyzed as described above. In controls, most blastomeres ($N = 9$) were uniformly A/G heterozygous (*LDLRAP1*$^{WT/Mut}$), as expected. However, a few blastomeres (6/15, 40.0%) demonstrated ambiguous sequencing and lost heterozygosity showing only WT or mutant allele (Supplementary Data 2).

Among 149 blastomeres recovered from injected embryos, majority (46.3%) showed heterozygosity with intact WT and mutant alleles and were deemed as non-targeted *LDLRAP1*$^{WT/Mut}$ (Fig. 1c and Supplementary Data 2). Among targeted blastomeres,

43/149 (28.9%) lost the mutant paternal A locus and showed WT G allele only (designated as *LDLRAP1*$^{homo-WT}$). Conversely, 14/149 (9.4%) blastomeres lost the WT allele but showed mutant A allele (*LDLRAP1*$^{homo-Mut}$). In addition, 2/149 (1.3%) blastomeres showed the presence of only one indel mutation (designated as *LDLRAP1*$^{homo-Indel}$). Other targeted blastomeres (14.0%) were heterozygous carrying indels at one or both alleles (*LDLRAP1*$^{WT/Indel}$, *LDLRAP1*$^{Mut/Indel}$ or *LDLRAP1*$^{Indel/Indel}$; Fig. 1c). There was no evidence of HDR with ssODN. In summary, similar to the *MYBPC3*, biallelic DSBs induced at the heterozygous *LDLRAP1* locus produced LOH in large number of blastomeres (59/149; 39.6%; Fig. 1d).

In summary, repair of biallelic DSBs induced in pre-implantation human embryos leads to a variety of on-target alterations, including large deletions, small indels, and HDR.

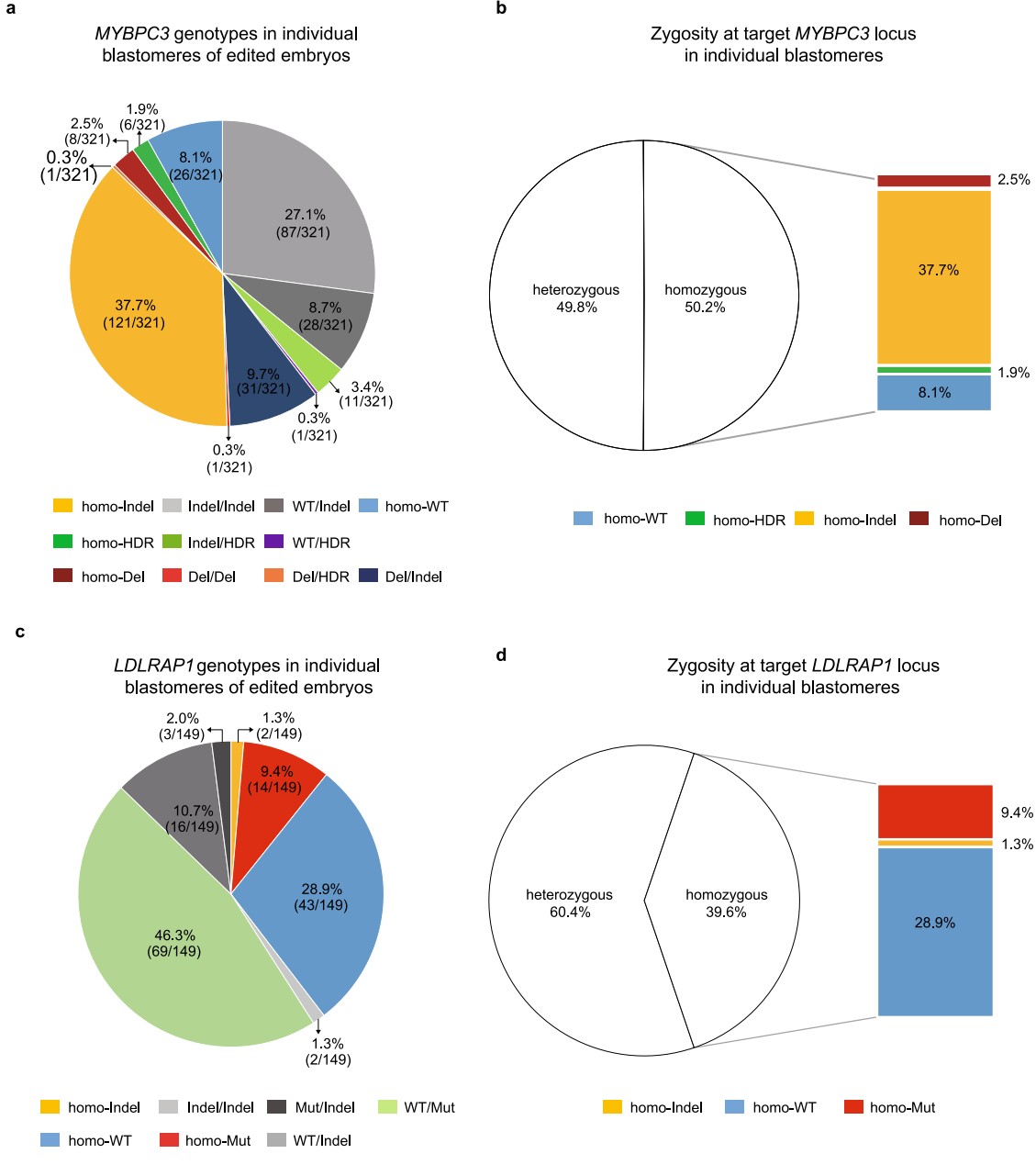

**Fig. 1 | Biallelic DNA DSB repair at *MYBPC3* locus of human embryos. a** On-target *MYBPC3* genotypes in individual blastomeres of human embryos injected with CRISPR/Cas9. **b** Zygosity at the target *MYBPC3* locus in individual blastomeres of human embryos injected with CRISPR/Cas9. Note that a large proportion of blastomeres carried homozygous genotype locus due to LOH. **c** On-target *LDLRAP1* genotypes in individual blastomeres of human embryos injected with CRISPR/Cas9. **d** Zygosity at the target *LDLRAP1* locus in individual blastomeres of human embryos injected with CRISPR/Cas9. A large proportion of blastomeres with homozygous genotype demonstrates LOH at the target locus. Source Data are provided as Supplementary Data 1 and 2.

However, targeted loci in a substantial portion of examined blastomeres exhibited LOH consisting of only one deletion or indel mutation. We reasoned that due to the random nature of NHEJ repair, the chances of generating identical deletions or indel mutations on both alleles are very low. It is possible that this phenomenon resulted from either presence of unidentified larger deletions or due to interhomolog gene conversion. Alternatively, ADO during WGA can also yield false-positive LOH or deletions as seen in few control samples. However, neither of these possibilities can be unequivocally excluded because these blastomeres cannot be reanalyzed.

## LOH in human embryos induced by monoallelic DSB

In an effort to further corroborate LOH, we induced monoallelic DSBs in heterozygous *MYH7* human embryos. We fertilized WT oocytes with sperm donated by a subject carrying a heterozygous mutation in exon 22 of *MYH7* gene (1 bp C > T substitution; g.15819 C > T, NG_007884.1) located on chromosome 14, and implicated in familial hypertrophic cardiomyopathy (HCM) (Supplementary Fig. 3b). We designed and selected the sgRNA selectively targeting only the mutant paternal *MYH7* allele and injected this sgRNA along with Cas9 protein and ssODN into cytoplasm of pronuclear stage zygotes 18 h after fertilization (Supplementary Fig. 4a). To differentiate from the WT sequence, ssODN template carried three synonymous single nucleotide substitutions. Injected zygotes ($N = 86$) along with non-injected controls ($N = 18$) were cultured for 3 days and then cleaving 4–8 cell stage embryos were disaggregated, and each blastomere was individually analyzed as described above.

As expected for heterozygous ($MYH7^{WT/Mut}$) sperm, on-target analysis of individual blastomeres disaggregated from 18 control embryos revealed that 9 were heterozygous mutant with most blastomeres (82.5%, 47/57) showing $MYH7^{WT/Mut}$ genotype (Supplementary Data 3). However, a few sister blastomeres (17.5%, 10/57) from these heterozygous embryos presented only one mutant or WT sequence likely due to ADO. Individual blastomeres ($N = 63$) from the remaining 9 controls presented only WT sequences, indicating that these embryos were uniformly homozygous ($MYH7^{homo-WT}$) as a result of fertilization with WT sperm (Supplementary Fig. 4b). Majority of injected embryos (58/86, 67.4%) were uniformly homozygous with each sister blastomere showing WT *MYH7* allele only ($MYH7^{homo-WT}$, Supplementary Fig. 4b, and Supplementary Data 4). While it is possible that these homozygous WT embryos originated from the WT sperm, increase in the portion of $MYH7^{homo-WT}$ embryos compared to controls was similar to our previous observations[6]. In injected group, 6/86 (7.0%) embryos were uniformly heterozygous (non-mosaic) carrying intact WT and indel mutation at or adjacent to the pre-existing mutant locus ($MYH7^{WT/Indel}$). The remaining embryos (22/86; 25.6%) were mosaic, each consisting of blastomeres with mixed $MYH7^{WT/Mut}$, $MYH7^{WT/Indel}$, $MYH7^{WT/Del}$, and $MYH7^{homo-WT}$ genotypes (Supplementary Fig. 4c). In contrast to *MYBPC3*, no evidence of HDR with ssODN was found in injected *MYH7* embryos.

Since at least one blastomere in mosaic embryos showed intact or edited g.15819 C > T locus, we presumed that these embryos were fertilized with the mutant sperm and used such mosaic embryos for on-target assessments. In-depth analysis of 134 blastomeres isolated from 22 mosaic embryos revealed that 14 (10.4%) were heterozygous with a WT and an intact mutant allele ($MYH7^{WT/Mut}$) while 66 (49.3%) were heterozygous with WT and indel mutations ($MYH7^{WT/Indel}$) and 4 (3%) carried 652bp deletion ($MYH7^{WT/Del}$) (Supplementary Fig. 4c). Consistent with our previous studies, the remaining blastomeres (50/134, 37.3%) lost the mutant allele and appeared as homozygous $MYH7^{homo-WT}$. Taken together, our results suggest that a large percentage of

monoallelic DSBs selectively induced at the mutant allele of heterozygous embryos induce LOH. As we argued above, LOH could be explained by either large deletions, gene conversion or allelic dropouts. Since DNA from these samples was already processed via WGA, we could not conclusively estimate the relative contribution of each of these possibilities.

## Allelic dropouts are common and may affect accurate genotyping in human preimplantation embryos

To estimate the rate of allelic dropouts in WGA DNA from single cell samples, we initially performed whole genome sequencing analysis of skin fibroblasts from a female proband volunteer and compared those to both her parents and cataloged all inherited heterozygous genomic variants. Based on this genetic information, we designed a custom sequencing panel targeting a total of 608 heterozygous loci scattered across all 23 chromosomes. When tested on a bulk DNA (without WGA) pooled from approximately 6 million fibroblasts from a proband, all 608 targeted regions were successfully amplified, and sequencing confirmed heterozygosity at all these loci suggesting 100% genotyping accuracy (Fig. 2a). We then used these fibroblasts synchronized at G0-G1 phase of the cell cycle for somatic cell nuclear transfer (SCNT) into enucleated oocytes and produced cleaving SCNT embryos[18]. We chose this approach because SCNT embryos in majority cases are genetically identical to donor fibroblasts, thus avoiding the limitations of assessing ADO in novel genomes of IVF embryos that are products of meiotic recombination during gametogenesis. We isolated 18 individual blastomeres from 11 SCNT embryos and 33 individual fibroblasts and subjected to single cell DNA WGA followed by sequencing of 608 targeted regions. Data analysis demonstrated that, on average, 84.8% of interrogated regions were amplified in each single cell sample and were suitable for genotyping. Among successfully amplified loci in embryonic blastomeres, on average 73.4% were heterozygous while remaining 26.6% loci appeared homozygous indicative of ADO (Fig. 2b; Supplementary Data 5). An average ADO rate in single fibroblasts was even higher (37.2%) likely due to lower DNA copy number in G1-arrested fibroblasts compared to cycling blastomeres. The range of ADO within each cell type varied significantly from 5 to 58.4% in blastomeres and from 3.1 to 75.7% in fibroblasts confirming previously reported frequencies[7,8]. These results confirm our concerns that a substantial number of actual heterozygous loci in embryonic samples may appear as false-positive homozygous due to WGA biases thus limiting the genotyping accuracy in human preimplantation embryos.

In attempt to find out if pooling of DNA from a few blastomeres can lower ADO occurrence, we pooled DNA from two ($N = 9$), three ($N = 8$) and six fibroblasts ($N = 3$) and processed for WGA. False-positive homozygosity was still detected in all three groups, but as expected the ADO rate was lower in the sample pooled from 6 cells (Fig. 2c; Supplementary Data 5). These data suggest that embryo biopsy consisting of several cells may produce more accurate genotyping. However, in research or clinical IVF settings when embryo mosaicism is expected, pooling is not desirable as it may mask existing genetic variations in individual cells.

Although genome-wide ADO rates show the frequency of genotyping artifacts per blastomere, in gene editing studies, the analysis of on-target modifications is based on the locus-specific data aggregated from multiple single-cell samples. Thus, evaluating locus-specific ADO may better reflect the characteristics of genomic regions under investigation and shed light on artificial LOH rates due to WGA biases. Analysis of heterozygosity among amplified genomic regions in 51 single cell samples revealed that an average ADO per locus varied from 6.8 to 83.3% with a mean rate of 31.8% (Fig. 2d and Supplementary Data 6). Data from the

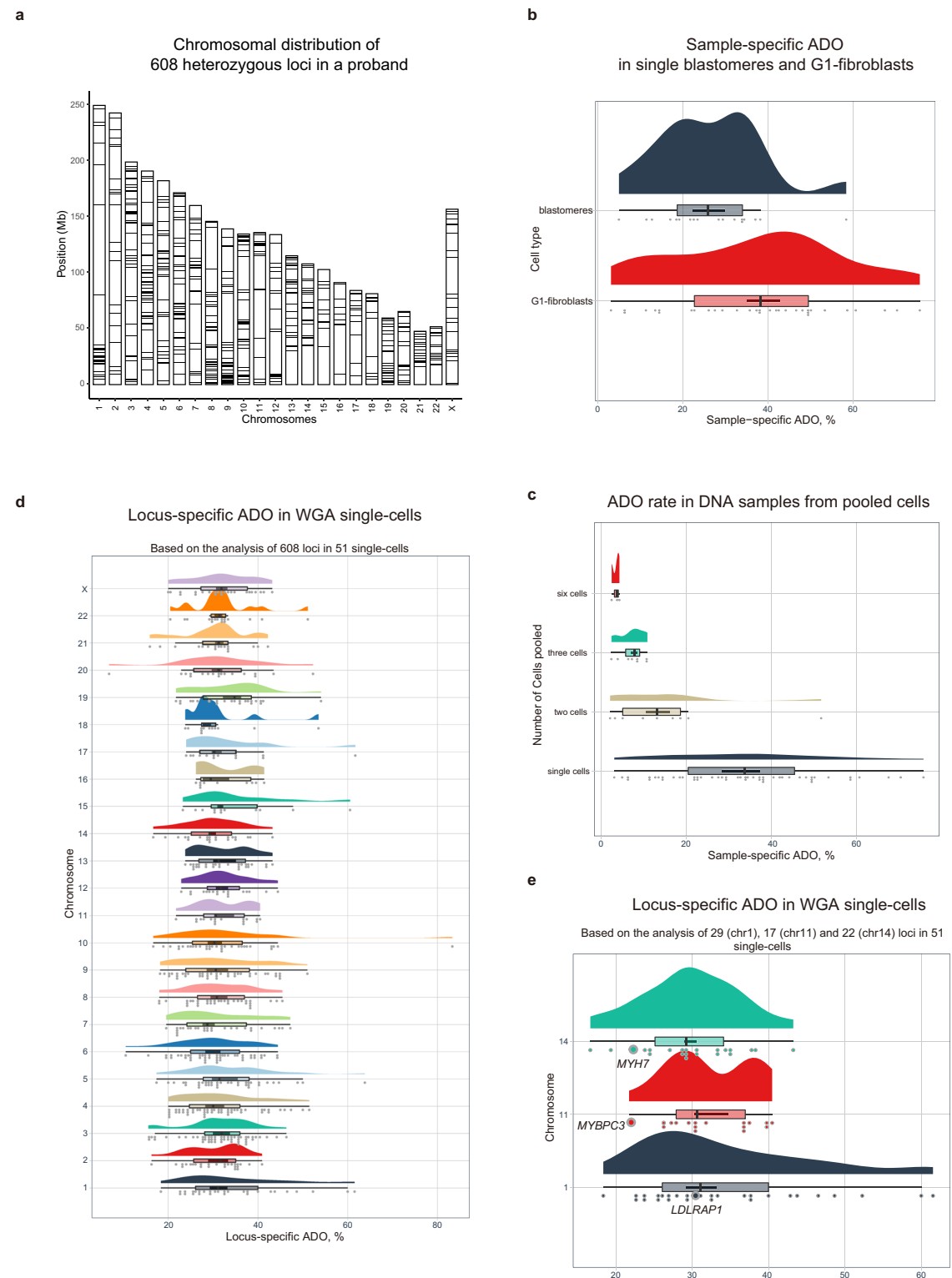

**Fig. 2 | Allelic dropouts due to whole-genome amplification biases limit the genotyping accuracy in human preimplantation embryos. a** Chromosomal distribution of 608 known heterozygous loci in female fibroblast donor used in the custom Ampliseq panel for investigation of ADO rates in single fibroblasts and blastomeres from SCNT embryos. **b** Distribution of sample-specific ADO rates observed in individual fibroblasts synchronized at G0–G1 phase of the cell cycle ($n = 31$) and blastomeres of cycling cleavage stage SCNT embryos ($n = 18$); $n$ represents biologically independent number of samples collected over four independent experiments. **c** Distribution of ADO rates in WGA single-cell DNA ($n = 51$) and WGA DNA samples pooled from two cells ($n = 9$), three cells ($n = 8$) and six cells

($n = 3$); $n$ represents biologically independent number of samples. **d** Chromosome- and locus-specific ADO rates for each of 608 targeted heterozygous loci. Based on the data from $n = 51$ single-cells examined over four independent experiments. **e** Frequency of ADO in three genomic regions closest to CRISPR/Cas9-targeted *MYBPC3*, *MYH7*, and *LDLRAP1* loci. Based on the data from 51 single-cells examined over four independent experiments. For each box plot in (**b**–**e**): center line, median; box bounds, 25th and 75th percentiles; whiskers, minimum to maximum within 1.5 interquartile range; data points outside whiskers, outliers. Source Data are provided as Supplementary Data 5 and 6.

amplicons closest to our three target loci uncovered similar frequency of ADO events in *MYBPC3* and *MYH7* regions (21.7 and 22.2% respectively) but higher for the *LDLRAP1* locus (31.1%) (Fig. 2e). These results indicate that in single cells, false-positive LOH in the targeted locus under investigation may differ not only from cell to cell but also from locus to locus due to individual characteristics of the region, making it difficult to account for ADO and complicating the comparative analysis of single-cell data in gene editing studies targeting different genomic regions.

## Validations of on-target modifications in ESCs derived from edited embryos

Bearing in mind of high ADO rates and to overcome the limitations of single blastomere analysis, we decided to corroborate on-target modifications seen in embryos in ESC lines established from edited blastocysts. To minimize number of human oocytes or embryos needed for deriving large number of ESC lines, we co-injected *MYBPC3* and *MYH7* sgRNAs along with Cas9 protein during fertilization of 62 WT oocytes with heterozygous (*MYH7^WT/Mut^*) sperm. Of these oocytes, 50 (80.6%) were fertilized and 29 (58.0%) zygotes reached the blastocyst stage (Supplementary Data 7). These fertilization and blastocyst development rates are within normal range for non-injected controls[6], suggesting that gene editing did not impede preimplantation development of human embryos. Next, we plated all 29 experimental blastocysts onto feeder layers that resulted in 14 primary ESC colonies. This high ESC derivation rate (48.3%; 14/29) is also within normal efficiency for non-edited embryos in our laboratory indicating absence of noticeable negative selection for gene edited ESCs[19]. Considering high frequency of mosaicism in gene edited embryos, we further dissociated each primary ESC colony into single cells and established a minimum of 10 individual subclones from each primary colony. The bulk ESC DNA isolated from approximately 5 million cells for each of 140 ESC clones without WGA was processed by the long-range PCR covering 8–10 Kb around both *MYBPC3* and *MYH7* cleavage sites. Amplification products then were processed by next-generation sequencing for simultaneous analysis of introduced short on-target DNA modifications as well as large deletions (Supplementary Fig. 5).

Based on on-target edits for the *MYBPC3* locus, all 10 sister subclones for each of 14 ESC lines carried identical genotypes, indicating that during ESC derivation, majority of mosaic variants were lost, likely due to clonal origin of ESCs (Supplementary Data 8). Analysis of *MYBPC3* locus in 140 individually sequenced ESC subclones, revealed that all 10 sister subclones from two ESC lines (ES-3 and ES-7) carried large on-target deletions of 1873 and 824 bp, respectively. The second allele in each sister subclone was presented as a small indel (*MYBPC3^Del/Indel^*) (Fig. 3a and Supplementary Data 8). All sister subclones for ES-12 and ES-13 were homozygous with 2 and 1 bp deletion, respectively, thus were genotyped as *MYBPC3^homo-Indel^*. In addition, all subclones for 4 ESC lines showed WT allele only and were designated as *MYBPC3^homo-WT^*. Remaining ESC lines were heterozygous and genotyped as either *MYBPC3^Indel/Indel^* or *MYBPC3^WT/Indel^* (Fig. 3a and Supplementary Data 8). These results suggest that similar to individual blastomeres, some ESC lines exhibited homozygosity consisting of identical indel mutations on both alleles. While frequency of such homozygosity in ESCs was lower than in embryos, we reasoned that since ESC DNA was not pre-amplified with WGA, ADO due to WGA biases or other technical artifacts can be excluded. We then screened all ESC clones for the presence of large (up to 8–10 Kb) deletions using long-range PCR products as described above for blastomeres. No additional deletions were found except those in ES-3 and ES-7. In addition, we investigated for the presence of very large deletions or complete loss of one chromosome by cytogenetic assay using G-banding karyotyping. Detailed G-banding analysis confirmed that all ESC lines carried normal euploid karyotypes without any

detectable large deletions or other cytogenetic abnormalities (Fig. 3b). Next, fluorescence in situ hybridization (FISH) assay was employed to detect and visualize the presence of *MYPBC3* alleles within individual nuclei of plated ESCs. The FISH probe was designed to hybridize the 187,812 bp fragment covering entire *MYBPC3* gene. All cell lines including controls demonstrated the presence of two signals in each nucleus, ruling out the possibility of large deletions or chromosome losses (Fig. 3c). We also performed allelic copy number assessments by Droplet Digital PCR (ddPCR) and found no differences between control and edited *MYBPC3^homo-Indel^* ESC samples (Supplementary Fig. 6a). Based on these results, we concluded that ESC clones with *MYBPC3^homo-Indel^* genotypes were indeed homozygous with intact two alleles.

We then carried our similar analyses on the *MYH7* locus in the same ESC lines and its subclones. *MYH7* locus genotyping in 140 individually sequenced ESC subclones, demonstrated that majority cell lines (8/14, 57.1%) were uniformly homozygous with each subclone showing WT *MYH7* allele only and were designated as *MYH7^homo-WT^*. In addition, three ESC lines were heterozygous with WT and intact g.15819 C > T mutation (*MYH7^WT/Mut^*), while one was heterozygous carrying the WT and indel mutation (*MYH7^WT/Indel^*). The remaining two (14.3%) ESC lines (ES-13 and ES-14) were mosaic, each consisting of two different sister subclones carrying different edits (Fig. 4a and Supplementary Data 8). No large deletions were detected at the *MYH7* locus in edited ESCs. Detailed G-banding analysis confirmed that all ESC lines carried normal euploid karyotypes without any detectable deletions or other cytogenetic abnormalities (Fig. 4b). FISH assay was also employed to detect and visualize the presence of both *MYH7* alleles within individual nuclei and all ESC lines including *MYH7^homo-WT^* demonstrated the presence of two signals in each nucleus consistent with the conclusion that all cells indeed carry two intact alleles (Fig. 4c). Moreover, ddPCR assay indicated that *MYH7* copy number in all ESC lines with *MYH7^homo-WT^* was similar to controls (Supplementary Fig. 6b).

Comparative analysis of heterozygosity at the *MYBPC3* and *MYH7* locus in edited embryos and ESCs showed that frequency of LOH (homozygous indel genotypes) in ESC clones was lower than that seen in blastomeres (14.3% vs. 37.7%) (Fig. 4d). It is likely that some LOH in embryos is artificial due to faulty WGA. However, the frequency of LOH at *MYH7* locus (homo-WT) in ESCs was comparable to that in embryos (57.1% vs 67.4%) (Fig. 4d).

## LOH due to interallelic gene conversion

We postulated that during fertilization and zygotic stages, parental alleles may not be equally accessible for cleavage by CRISPR/Cas9. Such a scenario would lead to initial targeting one of the parental alleles and generating an indel mutation on the oocyte or sperm allele first. During consequent cell cycles, the second allele becomes available and cleaved by CRISPR/Cas9, but DSB repair would now be resolved by an alternative interhomolog repair mechanism known as interallelic gene conversion[6,13,14] leading to copying of the first indel mutation to the second allele. One of the hallmarks of gene conversion is acquisition of homozygosity or LOH beyond the target region since DSB locus and the adjacent sequence become identical to the template allele. Therefore, we hypothesized that these results may represent a sequential repair by NHEJ and gene conversion.

To test this assumption, we investigated possibility of LOH at flanking heterozygous regions to the target locus in *MYBPC3^homo-Indel^* ESC clones as a result of gene conversion. We also investigated which parental allele was used as template for gene conversion. Using whole-genome sequencing, we screened blood DNA from egg donor 1 and the sperm donor used to generate some embryos and ESCs and identified four informative genomic variants (#3, #5, #6, and #14) differentiating parental alleles in ESCs that were located at −2623 bp, −2551 bp, −2069 bp downstream and +2,963 bp upstream from the target locus

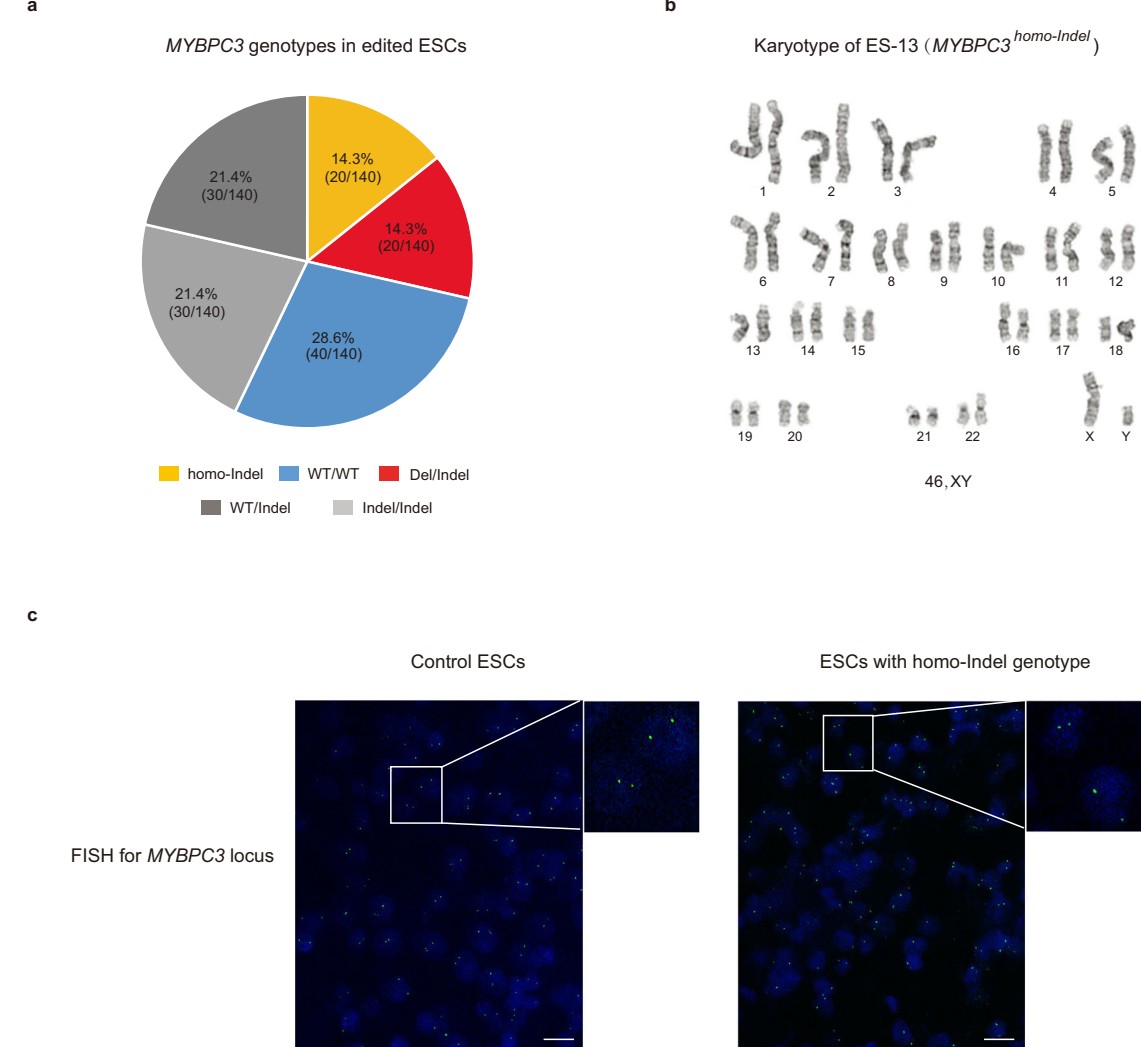

**Fig. 3 | Biallelic DSB repair at *MYBPC3* locus in human ESCs derived from edited embryos. a** *MYBPC3* genotypes in 14 ESC lines derived from CRISPR/Cas9 injected embryos. In contrast to embryos, a fewer ESC lines carried homo-indel genotypes. **b** G-banding analysis confirming that all ESC lines with *MYBPC3*^homo-Indel genotype exhibit normal diploid karyotypes without any detectable large deletions. **c** FISH labeling of the *MYBPC3* locus in ESC lines. Strong signals are seen on both chromosomes, indicating the presence of two intact *MYBPC3* alleles in all *MYBPC3*^homo-Indel cell lines (*n* = 3), *n* represents three biologically independent replicates of experiments. Bars 20 μm. Source Data are provided as Supplementary Data 8.

on chromosome 11 (Fig. 5 and Supplementary Data 9). In addition, 23 semi-informative heterozygous parental genomic variants scattered on both directions (−3324 bp to +16,185 bp) were also found to be useful to determine parental alleles. We initially interrogated these loci in heterozygous ESC clones produced from these gamete donors. As expected, all *MYBPC3*^Indel/Indel clones for ES-10 and ES-14 were heterozygous at genomic #3, #5, #6, and #14 positions carrying both parental variants, suggesting independent NHEJ repairs (Supplementary Data 9). Next, we genotyped these loci in all homozygous *MYBPC3*^homo-Indel subclones of ES-12 and ES-13 produced from these egg and sperm donors. All 10 clones of ES-12 with *MYBPC3*^homo-Indel genotypes showed a small region of homozygosity downstream of the target locus but were heterozygous at four informative genomic variants. Thus, these *MYBPC3*^homo-Indel did not carry LOH or LOH region could be short and undetectable for this parental combination (Fig. 5 and Supplementary Data 9).

However, all 10 clones of ES-13 with *MYBPC3*^homo-Indel were homozygous at all these four informative genomic variant positions, consistent with LOH (Fig. 5 and Supplementary Data 9). The actual region of homozygosity in ES-13 clones extended well beyond informative genomic variants loci and was also different among clones. For example, eight ES-13 clones were homozygous between genomic variants #2 and #26 positions (approximately 18.7 kb stretch), while remaining two clones were homozygous from genomic variants #2 to #17 (8.4 kb) (Fig. 5 and Supplementary Data 9). This indicates that ES-13 family consists of two different sister clones with various LOH length. Interestingly, all ES-13 clones carried homozygous maternal variants at all informative genomic variants positions but lost paternal variants suggesting gene conversion where maternal allele was copied to the paternal allele. Thus, the minimum conversion tract length for these clones was estimated at 7076 bp (from genome variants #3 to #15) but potentially could stretch up to 18,670 bp in length (from genome variants #2 to #26).

Postulating that acquisition of homozygosity or LOH are the hallmarks of gene conversion, we also genotyped homozygous *MYBPC3*^homo-WT clones (ES-11) derived from the same parental combination that were initially deemed as non-targeted. All clones were heterozygous at loci of informative genomic variants #3, #5, #6 and #14, indicating that these could be indeed an intact wildtype. However, a short region of homozygosity (from genomic variants #7 to #9;

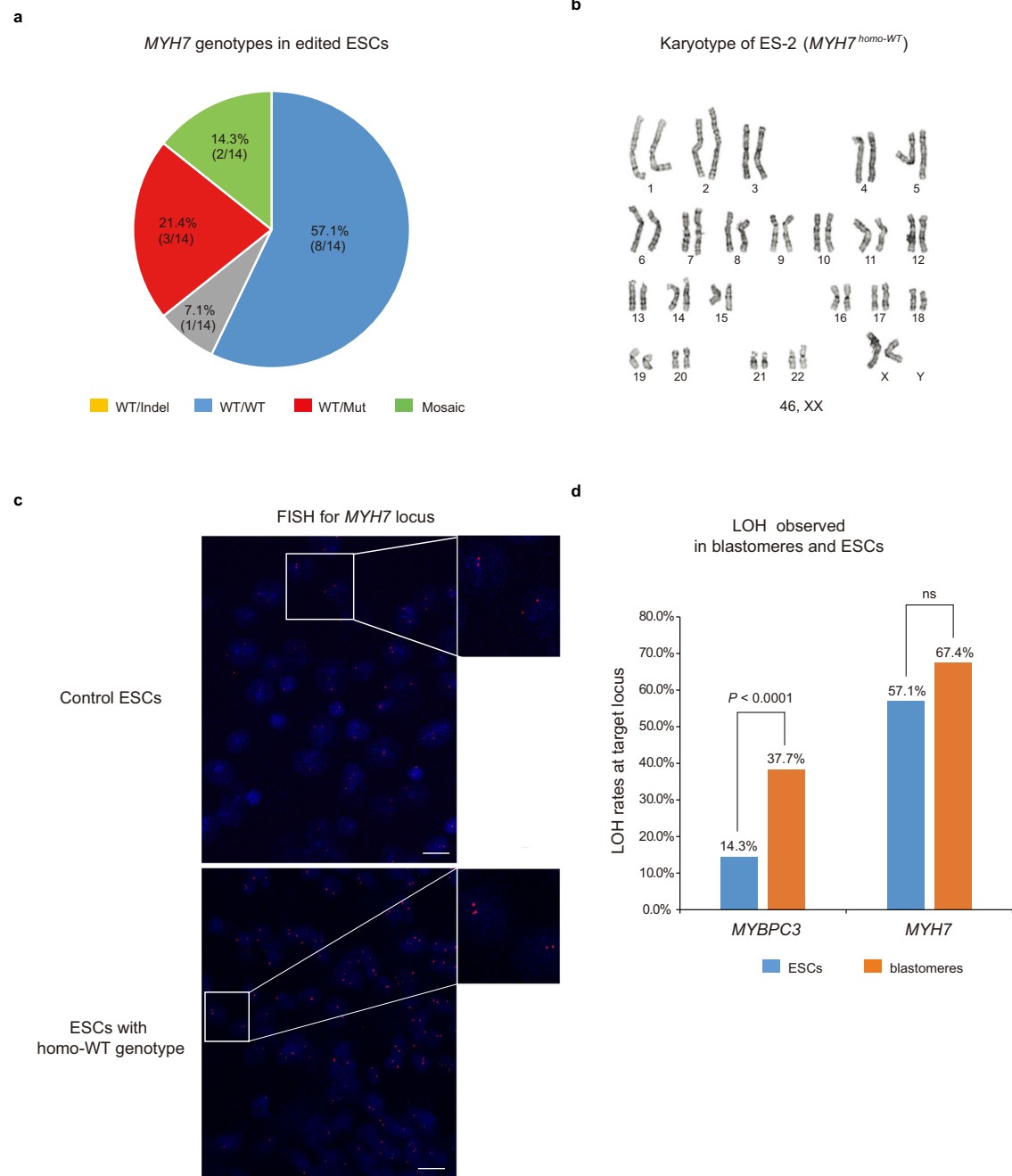

**Fig. 4 | Monoallelic DSB repair at the *MYH7* locus in ESCs derived from edited human embryos. a** *MYH7* genotypes in 14 ESC lines derived from CRISPR/Cas9 injected embryos. Similar to that seen in blastomeres, a substantial number of ESC lines carried *MYH7*^homo-WT genotypes. **b** G-banding analysis of *MYH7*^homo-WT ESCs derived from injected blastocysts exhibited normal euploid karyotype. **c** FISH labeling of the *MYH7* locus in *MYH7*^homo-WT ESC lines. Strong signals are seen on both parental chromosome 14, indicating the presence of two intact *MYH7* alleles ($n = 3$), $n$ represents biologically independent replicates of experiments. Bars 20 μm. **d** Comparative analysis of LOH in blastomeres and ESCs at the target *MYBPC3* and *MYH7* loci. Statistical significance tested with one-tailed Fisher's exact test (****$p < 0.0001$ for *MYBPC3*, $p = 0.0801$ for *MYH7*). Source Data are provided as Supplementary Data 8.

1,944 bp) downstream from the target locus was present, but since these genomic variants were heterozygous in egg and/or sperm donors, precise inheritance, in this case, could not be determined (Fig. 5 and Supplementary Data 9).

Due to a lack of egg donor genotype information for remaining ESC lines we could not determine the presence of LOH for ES-1, 5, and 6 families with *MYBPC3*^homo-WT genotype. However, when compared to the sperm and heterozygous sister ESC profiles, all 10 ES-1 clones exhibited a large region of homozygosity stretching to 7773 bp in length (Supplementary Fig. 7 and Supplementary Table 1).

In summary, our results confirm that *MYBPC3*^homo-Indel ESC clones display LOH adjacent to the target locus consistent with an interallelic gene conversion. This is likely facilitated by asynchronous induction of DSBs initially on the maternal allele leading to the generation of heterozygous indel mutation. Subsequent DSB on the paternal allele activates gene conversion resulting in copying of the maternal indel mutation and linked SNPs to the paternal allele. It is possible that sperm alleles are less accessible for DSBs likely due to chromatin compaction during early post-fertilization stages of development in human embryos.

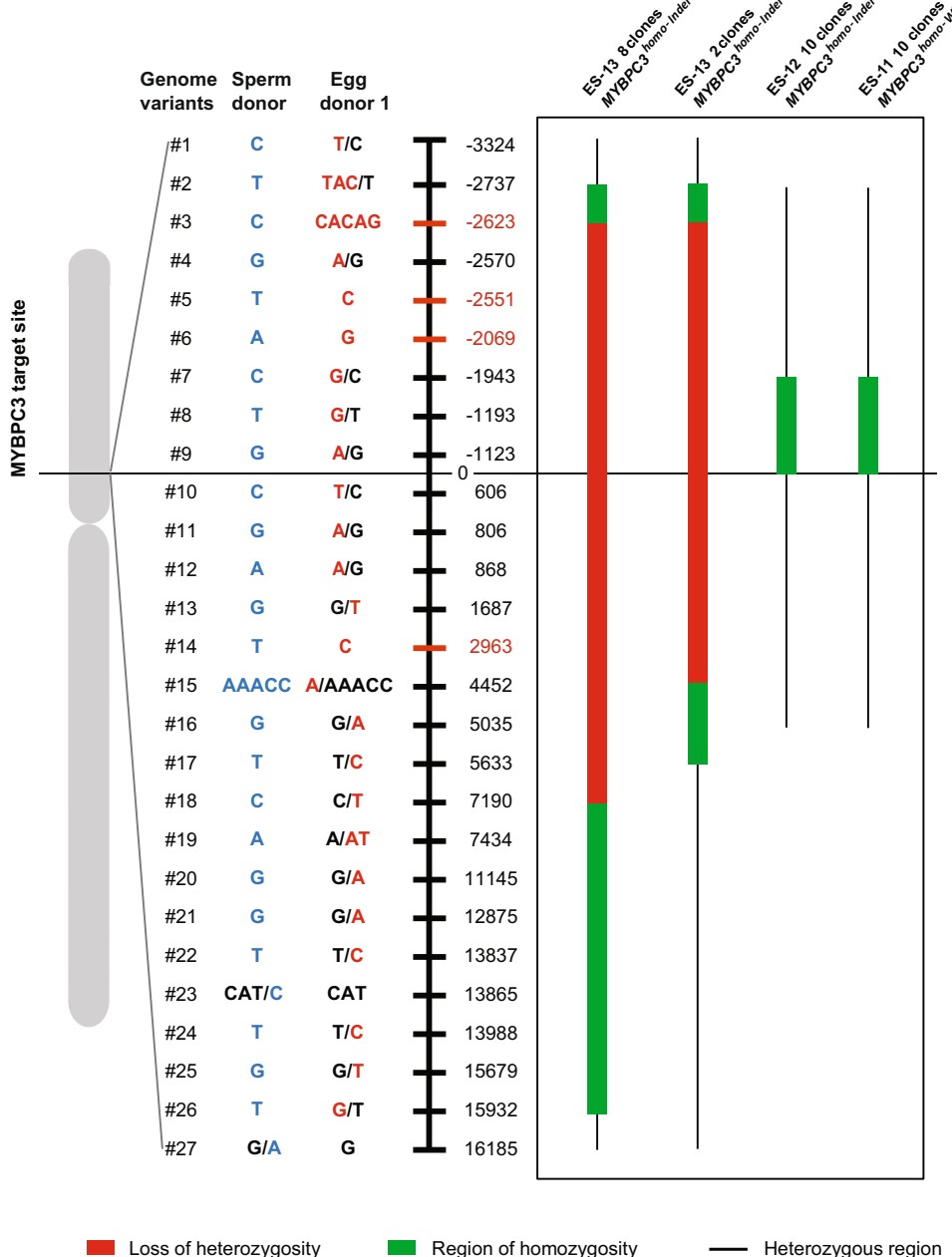

**Fig. 5 | LOH due to interallelic gene conversion in *MYBPC3^homo-Indel* ESC clones.** LOH induced by long gene conversion tracts. Schematic map of 27 informative genome variants or SNPs located upstream and downstream at various distances from the homozygous *MYBPC3* locus differentiating parental alleles in *MYBP3^homo-Indel* and *MYBPC3^homo-WT* ESC clones derived from edited embryos. LOH at each SNP site and genome variant was determined and compared to ESC clones with *MYBPC3^Indel/Indel* genotype from same egg and sperm donors. LOH and direction of conversion tract in each clone is indicated in color lines. Red lines indicate that these loci lost paternal nucleotides and become homozygous maternal. Green lines indicate that these loci are homozygous but LOH cannot be determined because of initial heterozygosity in contributing gametes. Black lines indicate that these loci are heterozygous. Source Data are provided as Supplementary Data 9.

## Discussion

Genetic analysis of preimplantation human embryos is complicated by minute amounts of genomic DNA available from biopsied samples or single cells. Thus, WGA is employed to produce higher DNA yields required for in depth sequencing analyses. However, applications of WGA are hampered by amplification biases, incomplete coverage of loci, and the small size of the DNA fragments leading to distorted sequence representation of the original template. Another potential source of errors involves collecting embryonic blastomeres soon after CRISPR/Cas9 injection, in which DSBs are not yet repaired, and thus cleaved fragments are not amplified and appear like partial loss of chromosome. In addition, due to high embryonic arrest, some

blastomeres of cleaving human embryos contain partially degraded DNA that may escape amplification. Hence, on- and off-target readouts of gene editing in human embryos must be validated to exclude false-positive DNA gains or losses. Leveraging repeated analysis of many single cells from the same somatic sample could minimize WGA artifacts but is not feasible for preimplantation embryos as each embryonic genome is unique. Here, we validated gene editing consequences seen in human preimplantation embryos in stable ESCs. ESCs are continuously growing cell lines that provide unlimited yield of high-quality DNA and live cells that can be subjected to in depth analyses of editing outcomes without WGA. The limitation of this approach is that ESCs represent a progeny of few epiblast cells,

indicating that majority of genetic variants of mosaic embryos may not be preserved.

Comparative analysis of on-target edits seen in single blastomeres and ESC clones showed obvious differences in frequency of some type of modifications. For example, the frequency of homozygous indel genotypes or LOH in ESC clones were substantially lower than that found in blastomeres. LOH could be explained by either large deletions, gene conversion or ADO. Some of these differences could be attributed to sampling differences but also could be due to artificial ADO during faulty WGA.

Analysis of ESCs demonstrates that DSB repair in human pre-implantation embryos produces an array of on-target modifications, including indel mutations, large deletions and gene conversions. Large deletions at the target locus ranging in size from 824 bp to 1873 bp were found in 14.3% of ESC lines when both alleles were targeted simultaneously (biallelic DSBs, *MYBPC3* locus) but not in cases when DSB was induced selectively on one allele (monoallelic DSBs, *MYH7* locus) suggesting that such deletions could be caused by simultaneous DSBs on both chromosomes.

We demonstrate that some ESCs indeed show LOH at the target region that is likely caused by interallelic gene conversion. Sequencing-based validation of gene conversion outcomes is difficult if parental alleles lack flanking informative SNPs or if the conversion tract is short. Indeed, our previous conclusions were challenged implying that the observed LOH in human embryos can also be interpreted as complete loss of a parental allele due to large deletions[14–16]. To address this issue, we derived stable ESC lines with LOH and applied FISH assay that provided visual, two signal confirmation that both alleles are intact. In addition, all ESC lines were karyotyped by G-banding to exclude aneuploidy. These results offer more conclusive evidence in support of gene conversion in human embryos.

Given that DNA DSB repair by gene conversion is a conserved mechanism across species[20], it is plausible that it might also act when programmable nucleases are introduced into non-human embryos. Indeed, a recent mouse study suggested that interhomolog repair occurs via gene conversion throughout the embryonic genome including pronuclear stage zygotes[13]. The strand exchange protein RAD51 was shown to significantly increase rates of Cas9-mediated gene conversion and produce homozygous knock-in of exogenous templates or to convert heterozygous alleles into homozygous[13]. In rats, allele-specific DSBs in heterozygous embryos were also repaired by an interallelic gene conversion at a frequency of 28% as judged by genetic and phenotypic analyses in live offspring[21]. This evidence from non-human species further supports our observations in ESCs that gene conversion is a repair outcome in response to DSBs induced in human embryos.

On the basis that parental genomes are physically separated into two pronuclei, possibility of inter-homolog gene conversion in zygotes was questioned[15]. However, recent imaging and sequencing studies in mouse zygotes demonstrated that DSB repair by interhomolog gene conversion could take place in late S/G2 stage zygotes[13]. As indicated above, while we introduced CRISPR/Cas9 into MII oocytes or pronuclear stage zygotes, repair of DSBs including by gene conversion could also occur later in cleaving embryos. Indeed, high incidences of mosaicism in gene edited cleaving embryos seen in our study suggest a considerable timelapse between CRISPR/Cas9 injections and DSB repair.

It is likely that gene conversion outcomes remain largely undetected in most gene editing studies[22]. Many animal studies reported cases of homozygous knock-out (identical indels) or knock-in (homozygous HDR)[23,24]. Based on our observations, some of these cases could be accounted for gene conversion. As discussed above, this can be only proven by the detection of LOH at flanking heterozygous loci. Mosaicism often masks gene conversion in pooled DNA samples, requiring single cell analysis.

Gene conversion could be applicable for future gene therapy to correct mutant alleles in heterozygous cells. To comply with strict requirements for germline gene therapy, gene conversion of heterozygous mutations back to the WT variants must be at much higher efficiency than observed in our recent study[6]. Extensive conversion tract and LOH up to 20 kb in size seen in some human ESCs in our study could be a safety concern. LOH could lead to uncovering of preexisting heterozygous variants on a template genome leading to homozygosity of deleterious alleles and disease in offspring. Moreover, gene conversion may also erase parent-specific epigenetic DNA modifications leading to imprinting abnormalities. Genome editing avoiding induction of DSBs could be more desirable and safer approach for germline gene therapy.

## Methods

### Study oversight

Guidelines, policies, and oversight defining research on human gametes and preimplantation embryos at Oregon Health & Science University (OHSU) were established by the Oregon Stem Cell and Embryo Research Oversight Committee (OSCRO). The studies were approved by the OHSU Institutional Review Board (IRB) and included independent review by the OHSU Innovative Research Advisory Panel (IRAP) and OHSU Scientific Review Committee (SRC). The approved studies were a subject for bi-annual external regulatory monitoring and Data Safety Monitoring Committee (DSMC) reviews.

### Ethics statement for research on human gametes and embryos

OSCRO established policy and procedural guidelines in 2008, formally defining the use of human embryos and their derivatives at OHSU, informed by the National Academy of Sciences' (NAS) Guidelines. These policies and guidelines permitted the procurement of gametes and embryos for research purposes, the creation of fertilized and SCNT human embryos specifically for research, genetic manipulation of human gametes and embryos, creation of human embryonic stem cell lines and molecular analyses. Together, OSCRO and the OHSU IRB worked concurrently to review and monitor applications for research studies involving human embryos at OHSU.

Human embryo and embryonic stem cell research policies and principles at OHSU were vetted over the course of a decade informed by the NAS guidelines, and subsequently affirmed by new guidelines released in 2015 by the Hinxton Group, the International Society for Stem Cell Research (ISSCR), and 2017 recommendations by the NAS and National Academy of Medicine joint panel on human genome editing. As a part of the review process, OHSU convened additional ad hoc committees to evaluate the scientific merit and ethical justification of the proposed study: the OHSU Innovative Research Advisory Panel (IRAP) and a Scientific Review Committee (SRC). Members of both committees were independent and their names were kept confidential from the research team; OHSU Research Integrity supervised all committee meetings, documentation, and formal recommendations.

IRAP Committee was tasked with deliberating ethical considerations related to using gene correction technology in human embryos for basic research at OHSU. The committee was composed of eleven members from internal and external sources: a lay member, a clinical ObGyn physician, three bioethicists, an OHSU Institutional Ethics committee member, three former OSCRO members, a clinical geneticist, and a clinician. Upon completion of the review, the IRAP recommended allowing this research "with significant oversight and continued dialog, the use of gene correction technologies in human embryos for the purpose of answering basic science questions needed to evaluate germline gene correction prior to the use in human models," at OHSU.

The established track record of the study team to uphold strict confidentiality and regulatory requirements paved the way for full OHSU IRB study approval in 2016, contingent upon strict continuing oversight, which includes: a phased scientific approach requiring evaluation of results on the safety and efficacy of germline gene correction in iPSCs before approving studies on human pre-implantation embryos; external bi-annual monitoring of all regulatory documents regarding human subjects; bi-annual Data Safety Monitoring Committee review; and annual continuing review by the OHSU IRB. The DSMC is required to remain active for the length of the approved IRB protocol and consists of four members: a lay member, an ethicist, a geneticist, and a reproductive endocrinologist. This committee conducts full review of all donations, the subsequent uses of these samples, and participant adverse events. The DSMC provides formal recommendations to the study team and IRB at the completion of each meeting.

### Informed consent
Written informed consent was obtained from all subjects prior to enrollment in the study. Study subjects included blood/skin, sperm, egg donors and women with infertility undergoing IVF willing to donate their discarded and/or excess gametes for this study. Subjects were informed of risks to participation, including risks associated with clinical procedures and loss of confidentiality. All consent forms included a lay language summary of germline gene modifications and SCNT, the ethical sensitivities surrounding germline gene modifications, and discussed the potential for incidental findings (genetic information potentially important to their future healthcare). Perspective participants were provided a copy of the consent form to review in advance of an in-person consent signing where the form is presented and discussed in further detail. Participants consented to the study, must be further consented if they wish to release their genetic data and/or materials to outside researchers.

As part of the formal consenting process, perspective donors are informed of risks to participation. The risk discussion covers in detail risks associated with clinical procedures and specifically highlights the risks to their genetic privacy; OHSU IRB has standard operating procedures, which are outlined in the consent form and study protocol to minimize a breach of confidentiality.

### Study participants
Adult skin, blood, and sperm donors were identified and enrolled in this study. In addition, healthy oocyte donors of 21–35 years of age were recruited locally, via print and web-based advertising. Cryopreserved, immature, excess or discarded oocytes donated by patients undergoing IVF treatments were also used in this study. The sex of the embryos was not determined or taken into account as part of the study design.

### Compensation
Study participants providing gamete, skin, or blood donations specifically for this research received financial compensation for their time, effort, and discomfort associated with the donation process at rates similar to gamete donation for fertility purposes. Infertility patients undergoing IVF whom donated immature oocytes did not receive any financial compensation.

### Ovarian induction
Ovulation stimulation was managed by OHSU REI physicians and followed established standards of care using a combination of self-administered injectable gonadotropins following 3–4 weeks ovarian suppression with combined oral contraceptives. Study participants self-administered medications for 8–12 days; the starting Follicle Stimulating Hormone (FSH) dose was 75–125 IU/day human Menopausal

Gonadotropins (hMG) was adjusted per individual response using an established step-down regimen until the day of human chorionic gonadotropin (hCG) injection. Gnrh antagonist was administered when the lead follicle was 14 mm in size. Subjects underwent ultrasound monitoring, and blood draws for estradiol levels. hCG and/or Lupron was administered when two or more follicles measured >18 mm in diameter. Subjects underwent oocyte retrieval via transvaginal follicular aspiration 35 h after hCG.

### Sperm donation
Study subjects were provided an at home semen collection kit or collected their sample at OHSU REI clinic. Semen was washed, counted, and analyzed for volume, sperm count, motility, and morphology.

### Skin donation
Study subjects had skin biopsy and blood draws performed at OHSU REI clinic performed by clinic staff and physicians. A numbing agent was injected into the back the arm and a core punch biopsy was obtained, a band-aid applied to the wound, and were followed post biopsy to ensure proper healing of biopsy. The 1–3 mm skin punch was placed into sterile PBS, processed, and plated to establish fibroblast cell line. A 5 ml blood sample was drawn into a heparinized tube and used for DNA isolation.

### Human ESC derivation
Zona pellucidae from blastocysts were removed with 0.5% pronase (Sigma P8811) and embryos were plated onto confluent feeder layers of mouse embryonic fibroblasts (mEF) and cultured for 6 days at 37 °C, 3% $CO_2$, 5% $O_2$ and 92% $N_2$ in ESC derivation medium. The medium consisted of DMEM/F12 (Gibco 11320-033) with 0.1 mM nonessential amino acids (Gibco 11140-050), 1mM L-glutamine (Gibco 21051-024), 0.1 mM β-mercaptoethanol (Sigma M6250), 5 ng/ml basic fibroblast growth factor (bFGF, Sigma F-0291), 10 μM ROCK inhibitor (Sigma SCM075), 10% fetal bovine serum (FBS, Hyclone Thermo Scientific SH30071.03) and 10% knockout serum replacement (KSR, Gibco 10828-028). ESC colonies were manually dissociated and replated onto fresh mEFs for further propagation and analyses. FBS and ROCK inhibitor were omitted after the first passage of ESCs and KSR was increased to 20%. All ESC lines have been authenticated by short tandem repeat (STR) genotyping, confirming their origin from the gamete donors from this study. ESC lines are available to researchers upon OHSU IRB approval and signed OHSU MTA.

### Fertilization and embryo culture
Mature MII oocytes were fertilized by intracytoplasmic sperm injection (ICSI) using fresh or frozen/thawed sperm as described earlier. Oocytes were placed into a 50 μL droplet of HTF (modified human tubal fluid) medium supplemented with 10% HEPES (Life Global #GMHH-50/100), overlaid with mineral oil (Sage IVF, Cooper Surgical ART-4008) and placed on the stage of an inverted microscope (Olympus IX71) equipped with a stage warmer and Narishige micromanipulators. A single sperm was drawn into ICSI micropipette and injected into the cytoplasm of each oocyte. Fertilized oocytes were then placed into dishes containing Global Medium (Life Global #LGGG-50/100) supplemented with 10% serum substitute supplement (Irvine Scientific 99193) and cultured at 37 °C in 6% $CO_2$, 5% $O_2$, and 89% $N_2$ in an embryoscope time-lapse incubator (Vitrolife). Successful fertilization was determined approximately 18 h after ICSI by noting the presence of two pronuclei and the second polar body extrusion.

### CRISPR/Cas9 design, selection and injection into human oocytes or zygotes
Multiple sgRNAs were designed for the *MYBPC3, MYH7 and LDLRAP1* locus and synthesized by in vitro transcription using T7 polymerase

(New England Biolabs). Each sgRNA along with Cas9 protein (PNA Bio CP01) and ssODN were transfected into blood or skin-derived iPSCs cells using Amaxa P3 Primary Cell 4D-Nucleofector Kit (Program). Three days after transfection, cells were harvested and DNA analyzed by targeted deep sequencing. CRISPR/Cas9 components that best performed in iPSCs were then selected for applications on human embryos. For the M-phase group, Cas9 protein (200 ng/μl), sgRNA (100 ng/μl) and ssODN (200 ng/μl) were co-injected with sperm into the cytoplasm of each MII oocyte during ICSI procedure as described before. For the S-phase group, the CRISPR/Cas9 components were injected into cytoplasm of pronuclear stage zygotes 18 h after ICSI.

sgRNA targeting:

MYBPC3(5′ GAGTTTGAGTGTGAAGTAT 3′)

LDLRAP1(5′ TGTGCTTGAAAACAGGAAGT 3′)

MYH7(5′ AAGTCCGAGGCTTGCCGCA 3′)

MYBPC3-ssODN (5′ AGATGGCCTCAGGGGAGCCAACCCTCATG CTCACCCTGCCTGGACAGAGCCCCCTGTGCTCATCACGCGCCCCTTG GAGGACCAGCTGGTGATGGTGGGGCAGCGGGTGGAGTTTGAGTGTG AAGTATCCGACGACGGCGCGCAAGTCAAATGGTGAGTTCCAGAAGC ACGGGGCATGGGTGTTGGGGGCATCTGCCCAG 3′)

LDLRAP1-ssODN(5′ AGGGAGCCAGGGGGCCTGGCCTGGAGGCCC CAGCCCTCCAGTGCAGACTTGCTCTGCCCTGGCTGACACTGCACCCC TCCCCATCCCCACTTAGTGTTTTCAGGCACAGGCTGTTACCCTC ACCGTAGCCCAGGCCTTCAAAGTCGCCTTTGAGTTTTGGCAGGTGT CCAAGGAAGGTGAGACTTTGCATCTACATTGTG 3′)

MYH7-ssODN(5′ TACTTCAAGATCAAGCCGCTGCTGAAGAGTGC AGAAAGAGAGAAGGAGATGGCCTCCATGAAGGAGGAGTTCACACG CCTCAAAGAGGCGCTAGAGAAGTCCGAGGCTCGACGGA AGGAGCTGGAGGAGAAGATGGTGTCCCTGCTGCAGGAGAAGAAT GACCTGCAGCTCCAAGTGCAGGCGGTGAGGCTCCTGGGCTA3′)

## Blastomere isolation and whole genome amplification

Injected oocytes or zygotes were cultured to the 4–8 cell stage and used for single blastomere analyzes. Briefly, the zona pellucida of cleaving embryos was removed using acid Tyrode's solution (NaCl 8 mg/ml, KCl 0.2 mg/ml, CaCl$_2$.2H$_2$O 2.4 mg/ml, MgCl$_2$.6H$_2$O 0.1 mg/ml, glucose 1 mg/ml, PVP 0.04 mg/ml). Embryos were then briefly exposed to a 0.05% trypsin solution, and individual blastomeres were mechanically separated using a micromanipulation pipette. Each blastomere was then placed into 0.2 ml PCR tube containing 4 μl PBS and stored at −80 °C. Whole genome amplification was performed using a REPLI-g Single Cell Kit (Qiagen 150345), according to the manufacturer's protocol. Briefly, frozen/thawed tubes containing blastomeres were treated with denaturation solution mix and incubated at 65 °C for 10 min. A master mix containing buffer and DNA polymerase was then added to each tube. The amplification reaction processed for 8 h at 30 °C in a PCR thermocycler. Whole genome amplification product was then diluted 100 times and used for downstream applications.

## Genotyping, sanger sequencing, and Long-range PCR

The target region for *MYBPC3* locus and SNP sites were amplified with PCR primers using PCR platinum SuperMix High Fidelity Kit (Invitrogen 12532-016). PCR products were purified by ExoSAP-IT reagent (Affymetrix), single purify condition were: 5ul PCR product with 2ul of ExoSAP-IT reagent, 37 °C for 15 min then 80 °C for 15 min. Then purified PCR product were sequenced by Sanger and analyzed by SnapGene® Viewer. Long-range PCR amplifications were performed by using TaKaRa LA Taq DNA Polymerase (Clontech). MYBPC3-1742bp-F GGCGGCACAGAGGGGATT, MYBPC3-1742bp-R TGGGACACCTT TATGCGGCT, MYBPC3-3054bp-F ACTCAGGGGTTGCTGAGAGA, MYBPC3-3054bp-R CGTCAATGGTCAGTTTGTGG, MYBPC3-8415bp-F CCAGGACAGCCACAAGGAAA, MYBPC3-8415bp-R ATCAGGTCG AAGTTCAGCCG.

## Donor fibroblasts preparation and somatic cell nuclear transfer (SCNT) procedures

Dermal fibroblasts from a proband female donor were cultured in 4-well dishes under standard conditions until they reach confluency. Confluent cells were synchronized in the G0/G1 phase of the cell cycle by culture in medium with low serum (DMEM/F12 medium with 0.5% FBS) for 2–4 days before SCNT. Enucleations, cell fusion, and artificial activations were performed. Briefly, meiotic metaphase II (MII) spindles were visualized under polarized microscopy and removed. Next, a disaggregated fibroblast was aspirated into a micropipette, exposed briefly to HVJ-E extract (Cosmo Bio LTD #ISK-CF-001-EX) and placed into the enucleated oocyte perivitelline space. After cell fusion, the SCNT oocytes were subjected to artificial activation.

## DNA fluorescence in situ hybridization (FISH)

FISH analyses were carried on both metaphase arrested and cycling interphase nuclei of ESCs. The probes were purchased from Empire Genomics, USA (Catalog # MYBPC3-20GR and MYH7-20-OR). FISH probes specific for *MYBPC3* (11p11.2 locus, -188Kb) were labeled using Green-dUTP, and for *MYH7* (14q11.2 locus, -177 Kb) were labeled using Orange-dUTP. Briefly, ESCs were treated with KaryoMAX Colcemide (Life Technologies) at a final concentration of 200 ng/mL for 1.5 h at 37 °C. Treated cells were then detached by 0.25% trypsin/EDTA and incubated in hypotonic 0.075 M KCL for 20 min. Cells were next fixed with methanol: acetic acid (3:1 v/v) and dropped onto a slide and dried on a hot plate at 60 °C. The samples were dehydrated using ethanol (70, 85, and 100%) for 1 min in each and dried in air. Slides were applied with the probe mixture, covered with an 18 mm$^2$ coverslip, and incubated in a humidified Thermobrite® system (Leica) set at 73 °C for 2 min, and then 37 °C for 16 h. The incubated slides were rinsed with washing solution 1 (0.3% Igepal/0.4 × SSC) and washing solution 2 (0.1% Igepal/2 × SSC). Slides were mounted in ProLong™ Gold Antifade Mountant with DAPI (Life Technologies) and observed using a fluorescence microscopy equipped with a cooled CCD camera. Images were captured and analyzed by ISIS analysis software (MetaSystem GmbH).

## Droplet digital PCR (ddPCR)

Copy numbers for *MYH7* and *MYBPC3* DNA were analyzed with the QX200 Droplet Digital PCR system (Bio-Rad). The ddPCR reaction mixture were prepared with "ddPCR Supermix for Probes (No dUTP)" (Bio-Rad 1863024) according to the manufacturer's protocol. The reaction mixture was then loaded into a disposable plastic box (Bio-Rad 1864008) with 70 μL droplet generating oil (Bio-Rad 1863005) and placed into the droplet generator (Bio-Rad). After droplet generation, the PCR amplification was run under the following cycling conditions: 1 cycle at 95 °C for 5 min, 45 cycles at 94 °C for 30 s, at 56 °C for 60 s, 1 cycle at 98 °C for 10 min, and holding at 4 °C. Finally, PCR results were analyzed by the droplet reader (Bio-Rad) and QuantaSoft Analysis Pro software (Bio-Rad v1.7.4.0917). Primer sequences for ddPCR were: MYBPC3-F AGCTCTTTGTGAAAGGTG, MYBPC3-R TCTGGAACTCAC-CATTTG; MYH7-F AGCTCTACTTCAAGATCAAG, MYH7-R CAGGT-CATTCTTCTCCTG. Probe sequences for ddPCR: MYBPC3-P FAM-CTCAAACTCCACCCGCTGCC-BHQ1, MYH7-P FAM-ACACCATC TTCTCCTCCAGC-BHQ1.

## Targeted deep sequencing analysis

To allow for both the on-target edits analysis and the detection of large deletions (up to 8 kb in size) in ESC clones, long-range PCR products targeting *MYBPC3* and *MYH7* loci were normalized to 0.2 ng/μl and 1 ng was used for library preparation with the Nextera XT DNA kit (Illumina) following manufacturer's instructions. Paired-end sequencing was performed on Illumina MiSeq platform as 2 × 250 bp at an average coverage depth of 3,698X. Sequencing reads underwent adaptor trimming using Trim Galore (v0.6.3) and were mapped to

chromosome 11 (*MYBPC3*) and/or 14 (*MYH7*) fasta sequence from GRCh38 genome build using BWA-MEM (v0.7.17). Duplicate reads were identified and marked by Picard tools (v2.18.2). SNPs and indels were discovered with FreeBayes (v1.3.1) with local left-alignment of indels and further normalized with SAMtools (v1.10). We further applied additional filters (QUAL > 1, SAF > 0, SAR > 0) to keep only high-confident variant calls. Large deletions were discovered with Delly (v0.8.7) and Lumpy (v0.3.1) using split-reads identification.

### Whole genome sequencing analysis

For analysis of LOH in ESCs and allelic dropouts in SCNT-embryos and donor fibroblasts, informative genomic variants were uncovered in gametes and fibroblast donors using whole-genome sequencing (WGS). WGS libraries were prepared using TruSeq Nano DNA Library Prep Kit (Illumina) following the manufacturer's instructions. Paired-end sequencing was performed on Illumina NovaSeq 6000 platform as 2 × 151 bp at an average coverage depth of 77.46X and uniformity of 96.13. Raw fastq files were uploaded to the Illumina BaseSpace Sequencing Hub for downstream processing. Genomic reads were aligned against GRCh38 human genome assembly and SNVs and indels were called using Dragen Germline Pipeline Version 3.6.3. For easier identification of informative heterozygous genomic variants and phasing, respective trio gVCFs were further subjected to Dragen Joint Genotyping Pipeline 3.6.3.

### Ampliseq library preparation and sequencing data analysis

DNA concentrations were normalized to 10 ng/µl and 50 ng was used as an input for library preparation with custom designed Ampliseq panel and the Ampliseq Library PLUS kit (Illumina) following the manufacturer's instructions. Paired-end sequencing was performed on Illumina MiSeq platform as 2 × 300 bp at an average coverage depth of 143×. Genome Analysis Toolkit (GATK) preprocessing steps were applied to raw sequencing reads to produce BAM ready for variant calling. These included generation of uBAM via FastqToSam (picard tools v2.26.9), marking adapters via MarkIlluminaAdapters, converting uBAM to fastq via SamToFastq, mapping to GRCh38 genome assembly using BWA-MEM (v0.7.17) and merging BAM and uBAM with Merge-BamAlignment. Duplicate reads were retained. Genomic variants were then discovered with FreeBayes (v1.3.1) with local left-alignment of indels and further normalized with bcftools (v1.14). Additional set of filters (QUAL > 1 & QUAL/AO > 10 & SAF > 0 & SAR > 0 & RPR > 1 & RPL > 1) were applied to keep only high-confident calls. The resulting set of variant calling data was used for downstream analysis of allelic dropout rates in Rstudio (v1.3.1093), R (v4.0.3). Targeted loci with variant coverage depth of less than 12× were considered as not amplified and excluded from genotyping. Sample-specific allelic dropout rate was determined on a per cell basis as a fraction of homozygous loci out of total number of amplified targeted heterozygous regions (out of 608). Locus-specific allelic dropout rate was determined for each of targeted 608 genomic regions on a per locus basis using genotyping data from 51 single-cells.

### Statistics and reproducibility

No statistical method was used to predetermine the sample size. Instead, sample size was chosen based on similar previously published studies[6] and was reviewed by OHSU Data Safety Monitoring Committee and approved by OHSU Institutional Review Board. No data were excluded from the analyses. Oocytes and zygotes were randomly assigned to control and experimental treatments. The sequencing investigators were blinded to sample allocation during experiments and outcome assessment. Statistical analyses were performed using GraphPad Prism version 8. Statistical comparison of frequency data was done using one tail Fisher's exact test (Fig. 4d), where $P < 0.05$ was regarded as significant.

### Reporting summary

Further information on research design is available in the Nature Portfolio Reporting Summary linked to this article.

## Data availability

All data supporting the findings of this study are included in this published article and its supplementary information files. The uncropped gels are provided in the Source Data file. All data used to generate summary graphs and figures are provided in Supplementary Data Files and Supplementary Tables. The following publicly available datasets were used in this project: GRCh38 genome assembly [ftp. ensembl.org/pub/release-105/fasta/homo_sapiens/dna/Homo_ sapiens.GRCh38.dna.primary_assembly.fa.gz]. Raw amplicon sequencing data generated in this study has been deposited in the Sequence Read Archive of the NCBI under the BioProject ID PRJNA909213 [https://www.ncbi.nlm.nih.gov/bioproject/PRJNA909213], SRA ID [https://www.ncbi.nlm.nih.gov/sra/PRJNA909213] and are publicly available. Due to OHSU IRB regulations and Oregon laws as well as terms of the consents signed by research participants, we are unable to publicly upload and share whole-genome or whole-exome sequencing data outside of the OHSU network as these data sets can reveal the genetic identity of the study participants. Nevertheless, we are willing to share this information on a case-by-case basis, onsite at OHSU if approval is granted by OHSU Research Integrity and the IRB (Kara Drolet, Associate VP, ORIO irb@ohsu.edu). Additionally, the approved requestor will be directed to an OHSU compliance officer to initiate a Non-Disclosure Agreement. These measures are intended to ensure the confidentiality of our research participants while also striving for research transparency and reproducibility. Upon successful approvals the requestor will be escorted by a team member at all times and granted access to an OHSU computer in a shared office where they can access and review the data sets. All request should be initiated with OHSU Research Integrity and will move through their process, which may take upwards of 3–6 months to gain full access to all genome sequencing data that was generated in this work but not shared publicly. Source data are provided with this paper.

## Code availability

Whole-genome sequencing data analysis was performed using Illumina Dragen Germline Pipeline version 3.6.3 and targeted sequencing data analyses were performed with custom bash and R scripts based on previously published open-source software as described in the Methods section. Exact commands with parameters as well as other R scripts associated with generation of figures are available from the lead contact upon request, requests will be answered in 4 weeks.

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

## Acknowledgements

We thank the OHSU Institutional Review Board (IRB), innovative Research Advisory Panel (IRAP), Scientific Review Committee (SRC) and members of the Data Safety Monitoring Committee (DSMC), including Marci Messerle Forbes, Mark Bedau, Justin Courcelle, Lorna Marshall, and Barb Fisher for oversight and guidance of this study. We are grateful to all study participants for gamete and tissue donations and the Women's Health Research Unit staff, IVF laboratory staff, University Fertility Consultants, and the Reproductive Endocrinology and Infertility Division in the Department of Obstetrics and Gynecology, OHSU for their assistance in procurement of human gametes. We are grateful to Jun Wu and Juan Carlos Izpisua Belmonte for help with sgRNA selection. Studies conducted at the OHSU Center for Embryonic Cell and Gene Therapy were supported by OHSU institutional funds and a grant from the Burroughs Wellcome Fund. Studies conducted at Shandong university were supported by a grant from Chinese Thousand Talent Program. Studies conducted at Asan Medical Center and CHA University were supported by grants from National Research Foundation of Korea NRF-2015K1A4A3046807 and NRF-2020M3A9E4036527. Studies conducted at Anhui Medical University were supported by grants from the National Natural Science Foundation of China (82101802) and the Scientific Research of BSKY (XJ2020025) from Anhui Medical University.

## Author contributions

D.L., A.M., and S.M. conceived hypotheses and designed study protocols. A.K., D.B., T.O.L., S.A.K., D.M.L., D.H.W., P.B.D., S.K., S.B.H., and P.A. coordinated recruitment of gamete donors, conducted ovarian stimulations and oocyte retrievals. S.W.P. and J.S.K. designed and tested CRISPR/Cas9 constructs. N.M.G. and D.L. performed CRISPR/Cas9 injections, fertilization, embryo culture, blastomere isolation, and whole genomic amplifications. N.M.G. performed human SCNT. D.L., H.D., Y.Li, C.V.D., H.Y., T.C., H.Z., K.L.W., J.Y.Z., Z.Z.H., and Z.J.C. performed DNA extractions, PCR, and Sanger Sequencing. D.L., N.M.G., H.M., A.M., and R.T.-H. derived and cultured human ESCs. S.B.O. performed cytogenetic analyses. H.M., A.M., and H.D. performed targeted deep sequencing analysis. A.M. performed WGS and Ampliseq analysis. Y.Lee, S.J.S., J.S.H., J.P., C.J.K., and E.K. designed and performed FISH experiments. D.L. and Z.K. performed ddPCR. J.H.G., Y.L.Y., Y.G., and Y.S. performed WGS. D.L., A.M., and S.M. analyzed the data and wrote the manuscript. S.M. and P.A. supervised this project.

## Competing interests

The authors declare no competing interests.

## Additional information

[1]Center for Embryonic Cell and Gene Therapy, Oregon Health & Science University, 3303 S. Bond Avenue, Portland, OR 97239, USA. [2]Department of Obstetrics and Gynecology, the First Affiliated Hospital of Anhui Medical University, No 218 Jixi Road, Hefei 230022 Anhui, China. [3]Center for Reproductive Medicine, Shandong University, Jinan, Shandong 250001, China. [4]Department of Biomedical Science, College of Life Science and CHA Advanced Research Institute, CHA University, Seongnam, Gyeonggi 13488, Republic of Korea. [5]Stem Cell Center, Asan Institute for Life Sciences, Asan Medical Center, University of Ulsan College of Medicine, 88, Olympic-ro 43-gil, Songpa-gu, Seoul 05505, Republic of Korea. [6]Center for Genome Engineering, Institute for Basic Science, 70, Yuseong-daero 1689-gil, Yuseong-gu, Daejeon 34047, Republic of Korea. [7]Technology Center of Hefei Customs, Hefei 230009 Anhui, China. [8]Guangdong Provincial Key Laboratory of Genome Read and Write, BGI-Shenzhen 518083, China. [9]China National GeneBank, BGI-Shenzhen, Shenzhen 518120, China. [10]Department of Molecular and Medical Genetics and Knight Diagnostic Laboratories, Oregon Health & Science University, Portland, OR 97239, USA. [11]Institute of Neuroscience, State Key Laboratory of Neuroscience, Key Laboratory of Primate Neurobiology, CAS Center for Excellence in Brain Science and Intelligence Technology, Shanghai Research Center for Brain Science and Brain-Inspired Intelligence, Shanghai Institutes for Biological Sciences, Chinese Academy of Sciences, Shanghai 200031, China. [12]Division of Reproductive Endocrinology and Infertility, Department of Obstetrics and Gynecology, Oregon Health & Science University, 3303 Southwest, Bond Avenue, Portland, OR 97239, USA. [13]Knight Cardiovascular Institute, Oregon Health & Science University, 3181 Southwest, Sam Jackson Park Road, Portland, OR 97239, USA. [14]Department of Chemistry, Seoul National University, 599 Gwanak-ro, Gwanak-gu, Seoul 151-747, Republic of Korea. [15]Center for Reproductive Medicine, Ren Ji Hospital, School of Medicine, Shanghai Jiao Tong University, Shanghai Key Laboratory for Assisted Reproduction and Reproductive Genetics, Shanghai 200127, China. [16]Present address: School of Dentistry, Chonnam National University, 77 Yongbong-ro, Buk-gu, Gwangju 61186, Republic of Korea. [17]These authors contributed equally: Dan Liang, Aleksei Mikhalchenko. [18]These authors jointly supervised this work: Paula Amato, Shoukhrat Mitalipov. ✉e-mail: shoukhrat@gmail.com

