## [Peer Review File · Nature Communications]

Reviewers' Comments:

Reviewer #1:

Remarks to the Author:

In this manuscript, the Authors describe the spectrum of mutations arising at three distinct loci (MYBPC3, MYH7, and LDLRAP1) upon targeting them with CRISPR-Cas9 in human embryos and embryonic stem cells (ESCs). The Authors show that in vivo editing of these loci produces a broad range of alterations on both alleles, with a surprisingly low frequency of homology-directed repair.

The Authors compare mutations identified in zygotes and single blastomeres using whole-genome amplification (WGA) as opposed to mutations identified in ESC clones without using a WGA step and observe a higher frequency of loss-of-heterozygosity (LOH) events in the latter. Furthermore, by profiling single-nucleotide polymorphisms (SNPs) around the targeted regions, the Authors argue that some of the observed LOH events could be explained by a gene conversion mechanism.

This is an interesting study conducted on precious human samples, which unfortunately suffers from two main limitations: Firstly, the Authors' claim that WGA might be responsible for some of the alterations observed is not supported by a thorough comparison of different WGA methods, which are currently available and could be tested (e.g., MDA, MALBAC, LIANTI). Secondly, the number of loci examined (n=3) is very small, thus precluding the generalizability of the Authors' conclusions. In this context, the Authors could leverage WGA-free single-cell DNA-seq methods such as ACT (PMID: 33762732) to explore the effects of in vivo editing beyond the loci examined, for example to see whether large copy number alterations or mutations form in edited blastomeres. The Authors should also assess more quantitatively whether their ESC clones are truly genomically stable.

MAJOR REMARKS

- 1) The Authors claim that whole-genome amplification (WGA) can explain the higher burden of alterations detected in zygotes and single blastomeres as opposed to ESC clones not subjected to WGA. However, the Authors only tested one WGA method (Repli-g kit from Qiagen that is based on multiple displacement amplification (MDA)) without even describing it and without explaining why they chose this method in the Main section. The Authors should compare different WGA methods (MDA, MALBAC, LIANTI) on zygotes, single blastomeres and single ESCs, before they can conclusively distinguish between genome editing effects and WGA artefacts.
- 2) The approach used to prepare the sequencing libraries (tagmentation of PCR products) is only briefly described in the Methods section and there is no description of which of the three MYBPC3 PCR products shown in Supplementary Fig. 1b was used and whether a similar PCR strategy was also used for the other two loci examined. Why didn't the Authors opt for multiple shorter amplicons, adding Illumina adapters directly during the PCR, and instead chose transposition? The Authors need to motivate and explain their approach much more clearly in the Main section and also provide a detailed description of which amplicon was used for tagmentation for each of the three loci examined. What was the breadth and depth of coverage of each amplicon? Can the Authors provide IGV or equivalent genome browser views of the regions sequenced?
- 3) There is no visualization of the different types of mutations identified. It would be very useful if the Authors showed alignments of different editing outcomes to the corresponding targeted sequences.
- 4) The Authors state that the MYBPC3 and LDLRAP1 loci are homozygous while the MYH7 locus is heterozygous in the sperm and oocyte donors used to derive the embryos analyzed. However, there is no evidence supporting this statement: can the Authors show a Sanger sequencing chromatogram for these loci in genomic DNA extracted from peripheral blood of the donors? Related to this, on pg. 7 the Authors write: 'Remarkably, most homozygosity was due to LOH'. If I understand correctly, the Authors here refer to indels and the finding of indels on both maternal and paternal alleles (by the way, how do they distinguish between the two? There is no mentioning of allele phasing in the Main text nor in the Methods. Do the Authors account for PCR duplicates in

their analysis? If so, how? Do they have UMIs in their Nextera adapters?). This is a bit confusing because before the Authors state that this locus was homozygous to start with. Perhaps, the Authors could use an expression such as 'indel homozygosity'?

5) Why did the Authors choose to target MYBPC3? Is this gene and the locus targeted clinically relevant as in the case of LDLRAP1? The Authors should motivate more clearly in the Main text why they chose to target these three genes.

6) Pg. 7: why did the Authors target the LDLRAP1 locus immediately upstream of the heterozygous A/G SNP and not the SNP itself to test the efficiency of genotype correction? Related to this, if the sperm donor was A/A and the egg donors were G/G, the embryos should be heterozygous A/G, while on pg. 7 the Authors state that they induced breaks at the 'homozygous wildtype LDLRAP1 locus'. This is very confusing, please clarify.

7) Pg. 8: 'In summary, targeting the homozygous LDLRAP1 locus resulted in low editing outcomes'. Again, this is confusing because the locus should be heterozygous for the A/G SNP based on what the Authors describe. What is the frequency of A/G to A/A editing in vivo by using a ssODN carrying the wildtype A allele?

8) Pg. 10: 'In contrast to MYBPC3 locus, no HDR with ssODN was found in blastomeres ...': does it mean that there was no successful editing that would be of clinical relevance (i.e., correcting the disease-associated heterozygous mutation in exon 22 mentioned before on pg. 7)?

9) The Authors claim that the ESC cell lines that they derived were genomically stable based on karyotyping and DNA FISH for two (MYBPC3 and MYH7) of the three loci examined, however the data provided are scant and not quantitative. The Authors should show DNA copy number profiles for all the 14 ESC clones described in the manuscript as well as a thorough quantification of the DNA FISH experiments performed (e.g., histograms of the number of FISH signals detected per nucleus across at least 100 nuclei per ESC clone).

10) The number of loci examined is very small, precluding the generalizability of the results. Although this Reviewer is well aware that testing more loci would be a major effort that clearly goes beyond the scope of this study, the Authors could attempt to provide a more in-depth characterization of the effects of genome editing beyond the targeted loci examined, at least in the ESC clones. For example, the Authors could leverage long-read sequencing technologies such as Oxford Nanopore to assess mutations and rearrangements across several kilobases encompassing the targeted loci using the approach described here: PMID: 32042167. Furthermore, the Authors could use the recently described WGA-free single-cell DNA-seq method, ACT (PMID: 33762732), to explore the effects of in vivo editing beyond the loci examined, by profiling genome-wide copy number changes in large numbers of individual blastomeres.

ADDITIONAL REMARKS

MAIN TEXT

--Page numbers are missing making it difficult for this Reviewer to point to typos or parts to be changed

--The Authors submitted a version of the manuscript still containing changes tracked in Word indicating a previous submission to a Cell Press journal. While this is certainly not a big issue, it conveys the impression of a rushed work. Related to this, Highlights and STAR Methods are used in Cell Press journal articles, not in Nature Communications.

--Pg. 4: 'Typically, CRISPR/Cas9 is introduced': the adverb typically is inappropriate here because the Authors are describing a very specific application of CRISPR and not the typical/general/most frequent type of CRISPR experiment in cell lines.

--Pg. 4: spell out 'WGA' when it is used the first time (Abstract doesn't count, introduce acronyms again when using them for the first time in the Introduction/Results/Discussion).

--The text is often split in multiple small paragraphs that are logically connected and should be kept together. For example: pg. 5: 'Indeed, latest studies ...': this is a logical continuation of the previous paragraph, therefore it should be kept together.

--The Authors frequently skip the use of the article 'the': for example: pg. 4: 'frequency of HDR is lower' should be 'the frequency of HDR is lower'; pg. 5: 'and overcome limitations of analyzing' >> 'overcome the limitations of analyzing'. The Authors could consider a native English speaker to correct these and other minor grammatical errors throughout the manuscript. (Disclaimer: this Reviewer is not a native English speaker).

--Pg. 9: 'relative contribution of each these possibility' should be 'relative contribution of each of these possibilities'.

--Pg. 9: please consider using active form: 'Next, all 29 experimental blastocysts were plated' >> 'Next, we plated all 29 ...'; 'each primary ESC colony was further dissociated' >> 'we further dissociated each primary ESC colony ...'; etc.

--Pg. 12: 'Analysis of MYBPC3 locus in 140 individually sequenced ESC subclones': Fig. 2a shows 128.6% of wt/wt in 14 clones not 140. Please clarify and correct accordingly.

--Pg. 13: since Fig. 3 is basically the same as Fig. 2 but for a different gene, consider starting the second paragraph like this: 'We then repeated the same analysis using embryos in which we targeted the MYH7 locus'.

--Pg. 13: 'Comparative analysis of MYBPC3 locus in edited embryos' >> 'Comparative analysis of the MYBPC3 and MYH7 loci in edited embryos'.

--Pg. 14: 'and adjacent area': 'and the adjacent sequence'?

--Pg. 14: 'We screened blood DNA ...': using whole-genome sequencing? Please clarify and also add detailed information in the Methods section.

--Pg. 14: 'In addition, 23-semi informative': why 'semi'? (also, it should read '23 semi-informative').

--Pg. 16: 'It is likely facilitated' >> 'This is likely facilitated'.

--Pg. 16: 'It is possible. that': please remove the full stop after possible.

--Pg. 16: 'sperm alleles are less accessible': due to higher chromatin compaction? Please clarify.

--Pg. 17: 'For example, frequency of homozygous indel genotypes [...] were substantially lower' >> 'For example, the frequency of homozygous indel genotypes [...] was substantially lower'.

METHODS

--Please provide ethical permit number(s) and a copy of the Informed Consent in the Supplementary Information file.

--'Human ESC derivation' section: please add the Celsius degree symbol after 37 (currently a square).

--DNA FISH section: were the BAC probes purchased from a commercial vendor? If so, please provide the company name and cat. no. In any case, please provide detailed information about the genomic coordinates of the regions targeted by the BAC probes (this can be included in a Supplementary Table).

DATA AVAILABILITY

Please add a data availability statement with a link to a public repository (e.g., SRA) containing all the sequencing data (FASTQ and/or BAM files) described in this study.

FIGURES

--Please add Figure number on top of each figure.

--Fig. 1a: show the same plot separately for S-phase and M-phase embryos

--Fig. 2b: please indicate the clone # to which this karyotype refers to

--Fig. 2c and 3c: please provide larger and better resolved images showing multiple nuclei, with magnification insets of individual nuclei. Add scale bars.

--Fig. 3d: please add y-axis title.

--Fig. 4: column 'Egg donor 1': what is the difference between red and black? Please also explain why some coordinate numbers on the right are in red.

--Suppl. Fig. 1a and c: it is not easy to match the ssODN sequence shown with the targeted sequences above. Also, what do the red underscores indicate? Please clarify.

--Suppl. Fig. 1b: why are different embryos (correct?) shown in red for PCR2 and PCR3? Why is there no gel for PCR1? What was the rationale of having three nested amplicons? Please explain also in the corresponding part in the Main text.

Reviewer #2:

Remarks to the Author:

This manuscript examines the repair outcomes of gene editing in human embryos. Although examining all repair outcomes is meaningful, the most important conclusion is that the data support the possibility of gene conversion in human embryos. This study is a continuation of their published work, and adds evidence to support their published work, for which alternative interpretations have been proposed. In general, a lot of data were collected to support their conclusion. However, the authors are required to address the following concerns:

1. In the opinion of this reviewer, the major contribution of this study is the demonstration of gene conversion (or the repair of the damaged allele using the intact allele as the template) by generating ES cell lines from gene edited embryos and observing gene conversion. This should be reflected in the title and the abstract. Currently the title is vague and could be changed to better reflect the paper's major contribution.

2. Whereas gene conversion was observed, the authors also observed large deletions. Thus both gene conversion and large deletions could explain the LOH observed in human embryo genome editing. The authors should emphasize this conclusion in their discussion.

3. The authors targeted three genes, MYBPC3, LDLRAP1 and MYH7. From Supplementary Fig1C, the G/A SNP was within the target site although several bps away from the predicted cleavage site. Since the eggs were from "G/G" donors and the sperms were from a "A/A" donor. Thus the embryos should be G/A genotype, which were not homozygous. Unless this reviewer did not understand Supplementary Fig1C correctly, the rationale of discussing MYBPC3 and LDLRAP1 editing under the title of "LOH in human embryos induced by DSBs at homozygous loci" is unclear. The authors should arrange this section differently and explain why LDLRAP1 with such SNPs was edited.

4. Cas9 RNPs were injected into MII oocytes or fertilized egg in this study. The authors are suggested to include a diagram to show the overall experimental scheme editing each gene, including the genotypes of the eggs and sperms, the targets of the sgRNA (targeting egg, sperm or both?).

5. Supplementary Fig1C, the PAM region was also included in the sgRNA sequence, this is incorrect. PAM should be excluded from the sgRNA sequence.

6. Targeting LDLRAP1 with the sgRNA sequence specific to the paternal allele, the authors observed loss of the paternal variant in some blastomeres (34.1%; 43/126), and loss of maternal alleles in 14/126 (11.1%) blastomeres. The rest were G/A genotypes. The authors should explain why this could happen. Was it caused by WGA loss of alleles? If yes, why the loss was unequal? One likely explanation is that the sgRNA was also able to cut the maternal allele (but with lower efficiency due to the single nt mismatch). The use of the unedited paternal allele as the template to repair the DSBs explained the loss of the maternal allele. The authors should perform experiments to check whether the sgRNA can cut the maternal allele and whether gene conversion can happen in G/A cell lines.

7. In Supplementary Fig.1C, the A/G polymorphism is certainly within the sgRNA target site. The statements that "The sgRNA designed to target the homozygous wildtype locus immediately upstream of the SNPA/G" and "All blastomeres (N=9) derived from control embryos were uniformly homozygous at the target site (LDLRAP1homo-WT) but heterozygous at the adjacent SNP locus (SNPA/G)" are misleading and confusing. These sentences should be changed to be accurate.

8. Line 155, "WT allele" should be clarified. Does it refer to "alleles without mutations caused by Cas9" or the allele with a G at the A/G SNP? The authors are suggested to distinguish the two regions by "cleavage site" (rather than target site, which in this reviewer's opinion, is the region complementary to the sgRNA and include the A/G SNP locus), and "the A/G SNP".

9. In Supplementary Fig.1C, the statement that the sgRNA targets "both wild type alleles" is

inaccurate. In each embryo, only one wild type allele is present (in this case, "wildtype" means the G SNP).

10. In Fig1C, 1D and related text, the authors are suggested to list both the types of mutation at the cleavage site and the A/G SNP, first maternal allele then paternal allele. For example, wt/Indel:G/A; Indel/Indel:A/A.

Minor concerns:

1. In Supplementary Fig.1A and 1C, the ssODN should be aligned to the reference sequence (with the sgRNA target site indicated) to clearly show the nucleotide changes.

2. Please check typo: "Large deletions at the target locus ranging in size from 145bp to 3.8kb were found in in 14.3% of"

Reviewer #3:

Remarks to the Author:

In this study, Dr Lang and colleagues aim to characterize the spectrum of on-target mutations induced by gene editing of human embryos at the MII or zygote stage by molecular analysis at different developmental stages, cleavage stage (individual blastomeres collected on 3 of embryo development) and bulk DNA of derived ESCs.

To overcome the limitation of single cell analysis or pooled blastomeres from a single embryo that would prevent mosaicism assessment, the authors validated on-target edits seen in human embryos by analysis of embryonic stem cells (ESCs) derived from targeted embryos that provided ample DNA for detailed sequencing. Moreover, ECS analysis would not require whole-genome amplification, avoiding the well-known amplification biases.

Comparative analysis of blastomeres and ESC demonstrated a more frequent loss of heterozygosity

(LOH) in embryos than in ESCs suggesting false-positive readouts due to WGA. Some of these differences could be attributed to sampling differences but also to the artificial loss of one allele during faulty WGA.

For example, the frequency of homozygous indel genotypes in ESC clones was substantially lower than that found in blastomeres. However, this wasn't true for all loci investigated. For instance, the authors reported that the frequency of LOH at the heterozygous MYH7 locus, (homo-WT) in ESC was comparable to embryos (57.1% vs 67.4%).

Analysis of ESCs demonstrates that DSB repair in human preimplantation embryos produces an array of on-target modifications including indel mutations, large deletions, and gene conversions. The authors attempted to exclude aneuploidies by FISH and G-banding on ESC showing LOH at target loci, supporting that authentic LOH is likely caused by interallelic gene conversion in human embryos.

Fertilization, blastocyst development rates and ESC derivation are within the normal range for non-injected controls, suggesting that gene editing did not impair the preimplantation development of human embryos and subsequent ESC derivation.

This is a very interesting and well-developed paper from leading authors in the field. Overall, I think the data and discussion are good and reasonable, but several assays are missing, and the authors should reformulate their conclusions and discussion. Their conclusions are too strong for the limited number of techniques that they use, and more caution is needed in their interpretations. For instance, they make conclusions about ADO based on one SNP (mentioned in line 144).

They focus on the introduction of ADO due to WGA, but I think they should delve deeper into

different types of ADO events, and at least mention that these may be different depending on the WGA technique for instance.

Although it is undoubtedly true that WGA (particularly MDA-based WGA) results in frequent ADO, this is usually random across the genome and does not impact genotyping-based copy number analysis because ADO events are usually not contiguous along chromosomal regions; they are usually interspaced. This is a very well-known feature in clinical PGT, where ADO usually involve one marker of those selected for linkage analysis. LOH of large and contiguous chromosomal regions is exceptionally rare following WGA.

The authors should comment on this. Also, was the karyotype of targeted embryos analyzed for aneuploidies? Did they check karyotypes of targeted blastomeres with PGTA?

I couldn't find this information.

The difference between LOH frequencies observed in blastomeres and ESC at the MYBPC3 locus and MYH7 locus doesn't seem to be statistically significant, given the small number of samples. Please quantify the statistical significance of this difference, for example providing a two-sample proportion test p-value (line 276,278)

In general, the authors mention a lot of numbers, but they do not always explain clearly where these are coming from. For instance, line 118, they mention 15 embryos, but which group do these belong to?

Considering mosaicism in gene-edited human embryos, a minimum of 10 clonally propagated ESC lines were established from each embryo. BUT hESC line might have arisen from a few cells not representative of the entire embryonic constitution. The limitation of this approach is that ESCs represent a progeny of a few epiblast cells, indicating that the majority of genetic variants of mosaic embryos may not be preserved. Can the authors comment on this?

Section "LOH in human embryos induced by DSBs at heterozygous loci"

- They utilize heterozygous sperm and therefore expect half of the embryos to originate from mutant sperm. For studying gene correction, it would be necessary to establish the true origin of the sperm for each embryo. It is always possible that a bias could be present (for instance, the mutant sperm may be more challenging to catch in a swim out).

- How did they track template used? Probably with the use of a synonymous SNP, but this information should be mentioned in the results.

-

- Line 215: A much clearer explanation for these conclusions is necessary. It seems that only the presence of the homo-WT allele was used to detect LOH, which is insufficient. If more techniques were used these should be clearly stated. Another explanation for the homo-WT allele could be the preferential occurrence of a specific indel, for instance, due to microhomology-mediated end joining (MMEJ).

Section "Validation of on-target modifications in ESCs derived from edited embryos"

- Line 224: The rationale for targeting two sites simultaneously is lacking. To my understanding, for one target they attempt to correct a mutation, while for the other target they attempt to just introduce indels at a homozygous wild-type allele. This should be clarified. Preferentially, these different events should have been investigated in different embryos and not in the same embryo. An explanation for this is missing (in their discussion).

-

- Line 231: I find the statement that they know that there was no negative selection too strong. The authors should make a comparison between the genetic events found in the stem cell lines and the embryos. If I understand correctly, only 57% (8/14) of the embryos displayed the WT allele which seems to suggest that theoretically only one embryo would be corrected. Again, the origin of the sperm (mutant or wild-type) is needed.

-

- Line 241-242: An explanation for what constitutes a small indel and what a large indel should be made.

-
- Line 250: How do they know with certainty that no ADO is present? ADO can also be due to the presence of an SNP in the primer-binding site, so a total exclusion of ADO is not possible.
Section "LOH due to interallelic gene conversion"
- Line 290: Why was this SNP assay not utilized for the embryo samples? The authors explain that they think that ADO can impede further LOH investigations, but this paper misses more proof for this.

Further comments

Line 375: To address the issue of large deletions, the authors utilized a FISH assay, however the resolution here is quite low (100-200 kb) therefore a lot of deletions, which they found with their long-range PCR assay in the embryos, would be missed. Therefore, a long-range PCR assay in stem cells would be more fitting. Why do they not utilize this technique for stem cells?

In the stem cell studies they perform karyotyping, but this information is missing in the embryo part.

Figure 1D: Another graph type with more detail would be more fitting.

Any difference in outcomes for immature, discarded and MII donated oocytes?

Fertilization rate, cleavage rate and embryo development metrics need to be reported in supplementary.

Point-by-point response to the reviewer's comments

We are thankful to all reviewers for the time and efforts dedicated on reviewing our manuscript and the constructive feedback provided to improve its current presentation. Please find uploaded manuscript that contains a significant amount of new data requested by the reviewers. We have also revised the entire manuscript text, tables and figures to address all the comments and revisions suggested and we hope now they all match with your expectations. Revisions in the text are highlighted with red fonts (additions) or strikeouts (deletions).

Reviewer #1 (Remarks to the Author):

In this manuscript, the Authors describe the spectrum of mutations arising at three distinct loci (MYBPC3, MYH7, and LDLRAP1) upon targeting them with CRISPR-Cas9 in human embryos and embryonic stem cells (ESCs). The Authors show that in vivo editing of these loci produces a broad range of alterations on both alleles, with a surprisingly low frequency of homology-directed repair.

The Authors compare mutations identified in zygotes and single blastomeres using whole-genome amplification (WGA) as opposed to mutations identified in ESC clones without using a WGA step and observe a higher frequency of loss-of-heterozygosity (LOH) events in the latter. Furthermore, by profiling single-nucleotide polymorphisms (SNPs) around the targeted regions, the Authors argue that some of the observed LOH events could be explained by a gene conversion mechanism.

This is an interesting study conducted on precious human samples, which unfortunately suffers from two main limitations: Firstly, the Authors' claim that WGA might be responsible for some of the alterations observed is not supported by a thorough comparison of different WGA methods, which are currently available and could be tested (e.g., MDA, MALBAC, LIANTI). Secondly, the number of loci examined (n=3) is very small, thus precluding the generalizability of the Authors' conclusions. In this context, the Authors could leverage WGA-free single-cell DNA-seq methods such as ACT (PMID: 33762732) to explore the effects of in vivo editing beyond the loci examined, for example to see whether large copy number alterations or mutations form in edited blastomeres. The Authors should also assess more quantitatively whether their ESC clones are truly genomically stable.

We appreciate the reviewer's time and efforts in evaluating our manuscript and useful suggestions to improve our presentation and conclusions. We agree that a comparison of different WGA methods could be useful to identify methods with the lowest possible allele dropout (ADO) rate. However, the utilization of precious and scarce human gametes and embryos for such experiments is complicated due to strict IRB restrictions

and ethical considerations. Such studies can certainly be conducted in embryos from model animals such as mice.

The goal of our study was to validate the LOH reported in gene edited human embryos by several recent studies (Alanis-Lobato et al., 2021; Ma et al., 2017; Papathanasiou et al., 2021; Zuccaro et al., 2020) that all used specifically Qiagen Repli-g WGA kit to pre-amplify DNA for further analyses. We now included additional studies and results in the revised manuscript clearly demonstrating that this WGA (Qiagen Repli-g) kit is prone to allelic dropouts. We interrogated here 608 heterozygous loci across all chromosomes preexisting in skin fibroblasts. We show that on average 27% of these loci appear as homozygous in individual blastomeres of human embryos generated by somatic cell nuclear transfer (SCNT). We chose the SCNT approach because donor fibroblasts can be sequenced for genomic variants prior to nuclear transfer and because SCNT embryos retain the original genotype of donor somatic cells. Conventional IVF platform complicates the assessment of preexisting parental genome variants in resulting embryos due to meiosis and recombination. We hope that adding this substantial amount of new data significantly improves the conclusions and impact of our study.

MAJOR REMARKS

1) The Authors claim that whole-genome amplification (WGA) can explain the higher burden of alterations detected in zygotes and single blastomeres as opposed to ESC clones not subjected to WGA. However, the Authors only tested one WGA method (Repli-g kit from Qiagen that is based on multiple displacement amplification (MDA)) without even describing it and without explaining why they chose this method in the Main section. The Authors should compare different WGA methods (MDA, MALBAC, LIANTI) on zygotes, single blastomeres and single ESCs, before they can conclusively distinguish between genome editing effects and WGA artefacts.

As we indicate above, we chose the MDA method, specifically Repli-g kit from Qiagen, to be consistent with our and several other studies (Zuccaro et al., 2020, Papathanasiou et al., 2021, Ma et al., 2017, Alanis-Lobato et al., 2020) that used this kit and reported LOH in gene edited human embryos. We revised the manuscript and included the reasoning behind using this WGA kit.

Additionally, we added new results on the analysis of 608 heterozygous loci across the human genome in non-gene-edited fibroblasts and somatic cell nuclear transfer (SCNT) embryos that show that WGA DNA from individual blastomeres and fibroblasts but not in from bulk fibroblast DNA (without WGA) produces artificial LOH due to allelic dropouts.

2) The approach used to prepare the sequencing libraries (tagmentation of PCR products) is only briefly described in the Methods section and there is no description of which of the three MYBPC3 PCR products shown in Supplementary Fig. 1b was used and whether a similar PCR strategy was also used for the other two loci examined. Why

didn't the Authors opt for multiple shorter amplicons, adding Illumina adapters directly during the PCR, and instead chose transposition? The Authors need to motivate and explain their approach much more clearly in the Main section and also provide a detailed description of which amplicon was used for tagmentation for each of the three loci examined. What was the breadth and depth of coverage of each amplicon? Can the Authors provide IGV or equivalent genome browser views of the regions sequenced?

We have added more details on the library preparation methodology to avoid confusion. PCR products illustrated in Supplementary Fig. 1b were not subjected to Illumina sequencing. Those were long-range PCR fragments from single blastomeres amplifying the MYBPC3 cleavage site in order to visualize gel bands and screen for the presence of large deletions. Targeted Deep Sequencing Analysis section in the Methods refers to the analysis of ESCs, not single blastomeres. Sequencing longer (up to 8-10Kb) ESC PCR products allowed us to screen for large deletions in situations when the exact coordinates of the deletion were unknown. Sequencing of multiple short amplicons, as suggested, would miss large deletions detected with our approach as DNA fragments carrying heterozygous large deletions would result in the amplification of only one allele without deletion. As stated in the Methods section, the average depth of coverage across samples with our deep sequencing approach was 3,698X. The breadth of coverage was 8.3Kb and 10Kb for MYBPC3 and MYH7 loci, respectively. As requested, we included IGV snapshots in the revised manuscript in Supplementary Fig. 5.

3) There is no visualization of the different types of mutations identified. It would be very useful if the Authors showed alignments of different editing outcomes to the corresponding targeted sequences.

We included in the revised manuscript detailed sequence information for each indel mutation in the Supplementary Table 1-4.

4) The Authors state that the MYBC3 and LDLRAP1 loci are homozygous while the MYH7 locus is heterozygous in the sperm and oocyte donors used to derive the embryos analyzed. However, there is no evidence supporting this statement: can the Authors show a Sanger sequencing chromatogram for these loci in genomic DNA extracted from peripheral blood of the donors? Related to this, on pg. 7 the Authors write: 'Remarkably, most homozygosity was due to LOH'. If I understand correctly, the Authors here refer to indels and the finding of indels on both maternal and paternal alleles (by the way, how do they distinguish between the two? There is no mentioning of allele phasing in the Main text nor in the Methods. Do the Authors account for PCR duplicates in their analysis? If so, how? Do they have UMIs in their Nextera adapters?). This is a bit confusing because before the Authors state that this locus was homozygous to start with. Perhaps, the Authors could use an expression such as 'indel homozygosity'?

We included Sanger Sequencing chromatograms from blood DNA of gamete donors showing zygosity in Supplementary Fig. 3. Yes, we refer to novel homozygosity when

only one indel mutation (or one large deletion, one ssODN) was seen by sequencing. We revised this sentence in the manuscript to avoid confusion. Since we Sanger sequenced DNA subjected to WGA, we could not distinguish if this was one copy indel or two identical on maternal and paternal alleles. As we reasoned, such novel homozygosity at the target locus could be caused either by large deletions, gene conversion, or ADO. However, we could not conclusively eliminate either of these possibilities in blastomeres. Therefore, we opted to continue the analysis of gene editing in stable ESCs.

5) Why did the Authors choose to target MYBPC3? Is this gene and the locus targeted clinically relevant as in the case of LDLRAP1? The Authors should motivate more clearly in the Main text why they chose to target these three genes.

We have previously targeted this *MYBPC3* locus on the paternal allele that carried pathogenic 4 bp deletion implicated in hypertrophic cardiomyopathy (Ma et al., 2017 and 2018). We have included the motivation in the revised text as to why we chose these loci.

6) Pg. 7: why did the Authors target the LDLRAP1 locus immediately upstream of the heterozygous A/G SNP and not the SNP itself to test the efficiency of genotype correction? Related to this, if the sperm donor was A/A and the egg donors were G/G, the embryos should be heterozygous A/G, while on pg. 7 the Authors state that they induced breaks at the 'homozygous wildtype LDLRAP1 locus'. This is very confusing, please clarify.

Our initial goal was to target specifically the mutant paternal A/A *LDLRAP1* locus but not maternal G/G, i.e induce monoallelic DSB as opposed to biallelic DSBs. However, neither of our designed and tested sgRNAs was specific to the mutant locus and rather cleaved both maternal and paternal alleles. Therefore, targeting of the *LDLRAP1* locus was considered as biallelic (similar to *MYBPC3*) even though the locus carried SNP. We agree that earlier description was confusing so we revised the text and figure.

7) Pg. 8: 'In summary, targeting the homozygous LDLRAP1 locus resulted in low editing outcomes'. Again, this is confusing because the locus should be heterozygous for the A/G SNP based on what the Authors describe. What is the frequency of A/G to A/A editing in vivo by using a ssODN carrying the wildtype A allele?

As mentioned above, we revised the text and figures to indicate that the locus was heterozygous but DSBs were induced at both alleles and is now referred to as biallelic cleavage to avoid confusion. No ssODN was found in edited blastomeres (N=149).

8) Pg. 10: 'In contrast to MYBPC3 locus, no HDR with ssODN was found in blastomeres ...': does it mean that there was no successful editing that would be of clinical relevance (i.e., correcting the disease-associated heterozygous mutation in exon 22 mentioned before on pg. 7)?

Yes, only indel mutations were detected but no evidence that ssODN was used as a template for HDR-based repair of *LDLRAP1* locus. We only detected ssODN-based repair when DSBs were induced at the *MYBPC3* locus but the frequency was very low (less than 5%) limiting clinical applications of HDR for mutation repair.

9) *The Authors claim that the ESC cell lines that they derived were genomically stable based on karyotyping and DNA FISH for two (MYBPC3 and MYH7) of the three loci examined, however the data provided are scant and not quantitative. The Authors should show DNA copy number profiles for all the 14 ESC clones described in the manuscript as well as a thorough quantification of the DNA FISH experiments performed (e.g., histograms of the number of FISH signals detected per nucleus across at least 100 nuclei per ESC clone).*

As suggested, we performed additional experiments and included new data showing quantitative FISH results. We performed Droplet Digital PCR (ddPCR) in edited ESC for copy number assay and provided these new results in the revised manuscript (Supplementary Fig. 6).

10) *The number of loci examined is very small, precluding the generalizability of the results. Although this Reviewer is well aware that testing more loci would be a major effort that clearly goes beyond the scope of this study, the Authors could attempt to provide a more in-depth characterization of the effects of genome editing beyond the targeted loci examined, at least in the ESC clones. For example, the Authors could leverage long-read sequencing technologies such as Oxford Nanopore to assess mutations and rearrangements across several kilobases encompassing the targeted loci using the approach described here: PMID: 32042167. Furthermore, the Authors could use the recently described WGA-free single-cell DNA-seq method, ACT (PMID: 33762732), to explore the effects of in vivo editing beyond the loci examined, by profiling genome-wide copy number changes in large numbers of individual blastomeres.*

We agree that off-target effects of gene editing in human embryos is important, however, the focus on this study was on-target effects and more specifically LOH.

We are currently working on a separate study to evaluate off-target consequences by whole genome sequencing of ESC lines derived from this project. We hope to provide these results in separate manuscript. We are definitely interested in testing novel WGA-free single-cell DNA-seq method and we thank the reviewer for these suggestions. As implementing this acoustic liquid handling technology would require significant investment of resources, we hope to try it in our future human embryo gene editing studies.

ADDITIONAL REMARKS

MAIN TEXT

--Page numbers are missing making it difficult for this Reviewer to point to typos or parts to be changed

We reformatted the manuscript and added page numbers.

--The Authors submitted a version of the manuscript still containing changes tracked in Word indicating a previous submission to a Cell Press journal. While this is certainly not a big issue, it conveys the impression of a rushed work. Related to this, Highlights and STAR Methods are used in Cell Press journal articles, not in Nature Communications.

Sorry for sloppiness, we submitted the revised manuscript that is formatted for Nature Communications.

--Pg. 4: 'Typically, CRISPR/Cas9 is introduced': the adverb typically is inappropriate here because the Authors are describing a very specific application of CRISPR and not the typical/general/most frequent type of CRISPR experiment in cell lines.

We revised this sentence to "In human embryos, CRISPR/Cas9 is frequently introduced".

--Pg. 4: spell out 'WGA' when it is used the first time (Abstract doesn't count, introduce acronyms again when using them for the first time in the Introduction/Results/Discussion).

In the revised manuscript, we spelled all acronyms when first time used in the main text.

--The text is often split in multiple small paragraphs that are logically connected and should be kept together. For example: pg. 5: 'Indeed, latest studies ...': this is a logical continuation of the previous paragraph, therefore it should be kept together.

Yes, we appreciated your suggestions. We went through the manuscript text carefully and combined logically connected sentences into one paragraph.

--The Authors frequently skip the use of the article 'the': for example: pg. 4: 'frequency of HDR is lower' should be 'the frequency of HDR is lower; pg. 5: 'and overcome limitations of analyzing' >> 'overcome the limitations of analyzing'. The Authors could consider a native English speaker to correct these and other minor grammatical errors throughout the manuscript. (Disclaimer: this Reviewer is not a native English speaker).

Yes, we agree that the manuscript could use grammatical edits and we provided the version edited by native English co-authors.

--Pg. 9: 'relative contribution of each these possibility' should be 'relative contribution of each of these possibilities'.

We revised this sentence.

--Pg. 9: please consider using the active form: 'Next, all 29 experimental blastocysts were plated' >> 'Next, we plated all 29 ...'; 'each primary ESC colony was further dissociated' >> 'we further dissociated each primary ESC colony ...'; etc.

Changed to active voice.

--Pg. 12: 'Analysis of MYBPC3 locus in 140 individually sequenced ESC subclones': Fig. 2a shows 128.6% of wt/wt in 14 clones not 140. Please clarify and correct accordingly.

We corrected to 140 ESC subclones in the Figure.

--Pg. 13: since Fig. 3 is basically the same as Fig. 2 but for a different gene, consider starting the second paragraph like this: 'We then repeated the same analysis using embryos in which we targeted the MYH7 locus'.

We revised the text as suggested.

--Pg. 13: 'Comparative analysis of MYBPC3 locus in edited embryos' >> 'Comparative analysis of the MYBPC3 and MYH7 loci in edited embryos'.

We revised the text as suggested.

--Pg. 14: 'and adjacent area': 'and the adjacent sequence'?

We revised the text as suggested.

--Pg. 14: 'We screened blood DNA ...': using whole-genome sequencing? Please clarify and also add detailed information in the Methods section.

We clarified in the text and added details in the Methods section.

--Pg. 14: 'In addition, 23-semi informative': why 'semi'? (also, it should read '23 semi-informative').

We revised the sentence as suggested. Semi-informative is referred to loci when gamete donors carry for example G/A and T/A SNPs. If an embryo inherits unique (G or T) alleles and presents as G/A, T/A, or G/T, we can clearly distinguish each parental allele. However, if an embryo inherits identical alleles and presents as A/A, this SNP becomes non-informative for parentage analysis.

--Pg. 16: 'It is likely facilitated' >> 'This is likely facilitated'.

We revised the sentence as suggested.

--Pg. 16: *'It is possible. that': please remove the full stop after possible.*

We revised the sentence as suggested.

--Pg. 16: *'sperm alleles are less accessible': due to higher chromatin compaction? Please clarify.*

Yes, one possible explanation is that during early post-fertilization stages, sperm chromatin is more tightly condensed and protected from nucleases by protamines compared to the oocyte genome. We revised the sentence to "likely" due to chromatin compaction'.

--Pg. 17: *'For example, frequency of homozygous indel genotypes [...] were substantially lower' >> 'For example, the frequency of homozygous indel genotypes [...] was substantially lower'.*

We revised this sentence as suggested.

METHODS

--Please provide ethical permit number(s) and a copy of the Informed Consent in the Supplementary Information file.

We provided copies of IRB approval and Informed consent forms in the manuscript files.

--*'Human ESC derivation' section: please add the Celsius degree symbol after 37 (currently a square).*

We added missing Celsius symbol.

--*DNA FISH section: were the BAC probes purchased from a commercial vendor? If so, please provide the company name and cat. no. In any case, please provide detailed information about the genomic coordinates of the regions targeted by the BAC probes (this can be included in a Supplementary Table).*

We added FISH probe details in the Methods section.

DATA AVAILABILITY

Please add a data availability statement with a link to a public repository (e.g., SRA) containing all the sequencing data (FASTQ and/or BAM files) described in this study.

We uploaded raw sequencing data in the public repository and provided a link. Our IRB regulations and Oregon laws prohibit disclosing genetic information of study participants that can reveal their identity. Since full WGS or WES datasets can be used to trace down the genetic identity of the study volunteers, we could not share this information publicly. However, partial sequencing datasets are deposited in the public repository.

FIGURES

--Please add Figure number on top of each figure.

We added Figure numbers on the top of each Figure.

--Fig. 1a: show the same plot separately for S-phase and M-phase embryos

We added genotype information for each blastomere for both the S-phase and M-phase groups in Supplementary Table 1. However, the on-target result of S-phase and M-phase groups show similarity in LOH and therefore, we put them together in Fig 1a.

--Fig. 2b: please indicate the clone # to which this karyotype refers to

Yes, we added these details in the Figure as suggested.

--Fig. 2c and 3c: please provide larger and better resolved images showing multiple nuclei, with magnification insets of individual nuclei. Add scale bars.

We provided new FISH images as suggested.

--Fig. 3d: please add y-axis title.

We added the y-axis title in Figure.

--Fig. 4: column 'Egg donor 1': what is the difference between red and black? Please also explain why some coordinate numbers on the right are in red.

Red font indicates maternal-specific nucleotides and black font represent paternal (sperm) nucleotides as the sperm donor. Coordinate numbers on the right in red or black indicate that in ESCs these loci become homozygous with only one nucleotide (maternal – red or paternal – black) present.

--Suppl. Fig. 1a and c: it is not easy to match the ssODN sequence shown with the targeted sequences above. Also, what do the red underscores indicate? Please clarify.

We aligned the ssODN sequence with the target region to make it more clearly. The red underlined nucleotides show substitutions in ssODN. We noted that in the figure legend.

--Suppl. Fig. 1b: why are different embryos (correct?) shown in red for PCR2 and PCR3? Why is there no gel for PCR1? What was the rationale of having three nested amplicons? Please explain also in the corresponding part in the Main text.

The blastomeres (shown in red fonts) carry a secondary band indicating large deletions. Long-range PCR amplicons were not nested. We designed 3 separate PCR pairs to screen independently for deletions of various lengths. No secondary band was detected in PCR1. Thus, we did not show gel for PCR1.

Reviewer #2 (Remarks to the Author):

This manuscript examines the repair outcomes of gene editing in human embryos. Although examining all repair outcomes is meaningful, the most important conclusion is that the data support the possibility of gene conversion in human embryos. This study is a continuation of their published work, and adds evidence to support their published work, for which alternative interpretations have been proposed. In general, a lot of data were collected to support their conclusion. However, the authors are required to address the following concerns:

1. In the opinion of this reviewer, the major contribution of this study is the demonstration of gene conversion (or the repair of the damaged allele using the intact allele as the template) by generating ES cell lines from gene edited embryos and observing gene conversion. This should be reflected in the title and the abstract. Currently the title is vague and could be changed to better reflect the paper's major contribution.

We appreciate the reviewer's time and efforts in evaluating our manuscript and useful suggestions to improve our results, conclusions, and presentation. We revised the title and abstract to reflect the main findings.

2. Whereas gene conversion was observed, the authors also observed large deletions. Thus both gene conversion and large deletions could explain the LOH observed in human embryo genome editing. The authors should emphasize this conclusion in their discussion.

We agree with this observation and emphasized this in the discussion as suggested.

3. The authors targeted three genes, MYBPC3, LDLRAP1 and MYH7. From Supplementary Fig1C, the G/A SNP was within the target site although several bps away from the predicted cleavage site. Since the eggs were from "G/G" donors and the sperms were from a "A/A" donor. Thus the embryos should be G/A genotype, which were not homozygous. Unless this reviewer did not understand Supplementary Fig1C correctly, the rationale of discussing MYBPC3 and LDLRAP1 editing under the title of "LOH in human embryos induced by DSBs at homozygous loci" is unclear. The authors should arrange this section differently and explain why LDLRAP1 with such SNPs was edited.

Sorry for the confusion. As we indicated above for reviewer 1 comments, it is indeed a heterozygous locus and our initial goal was to target specifically the mutant paternal A/A *LDLRAP1* locus but not maternal G/G, i.e induce monoallelic DSB as opposed to biallelic DSBs at both the *MYBPC3* loci. However, our several designed and tested sgRNAs for the mutant locus also mistargeted WT maternal allele. Therefore, targeting of the *LDLRAP1* locus was considered as biallelic (similar to *MYBPC3*) even though the locus carried SNP. We agree that designating this locus as "DSBs at homozygous

locus” is confusing and we revised text and subtitles to indicate that DSBs in case of *MYBPC3* and *LDLPA1* were “biallelic”. While in case of *MYH7* DSB was monoallelic.

4. Cas9 RNPs were injected into MII oocytes or fertilized egg in this study. The authors are suggested to include a diagram to show the overall experimental scheme editing each gene, including the genotypes of the eggs and sperms, the targets of the sgRNA (targeting egg, sperm or both?).

We included new diagram showing experimental design in the revised manuscript (Supplementary Fig.1).

5. Supplementary Fig1C, the PAM region was also included in the sgRNA sequence, this is incorrect. PAM should be excluded from the sgRNA sequence.

We corrected the Figure as suggested.

6. Targeting *LDLRAP1* with the sgRNA sequence specific to the paternal allele, the authors observed loss of the paternal variant in some blastomeres (34.1%; 43/126), and loss of maternal alleles in 14/126 (11.1%) blastomeres. The rest were G/A genotypes. The authors should explain why this could happen. Was it caused by WGA loss of alleles? If yes, why the loss was unequal? One likely explanation is that the sgRNA was also able to cut the maternal allele (but with lower efficiency due to the single nt mismatch). The use of the unedited paternal allele as the template to repair the DSBs explained the loss of the maternal allele. The authors should perform experiments to check whether the sgRNA can cut the maternal allele and whether gene conversion can happen in G/A cell lines.

We agree with this observation and as indicated above we could not design the sgRNA specific for the mutant paternal sequence. All tested sgRNAs cleaved both alleles. We agree that an increase in the loss of paternal variants vs. maternal could indicate skewed gene conversion due to more frequent DSBs on the paternal allele.

7. In Supplementary Fig.1C, the A/G polymorphism is certainly within the sgRNA target site. The statements that “The sgRNA designed to target the homozygous wildtype locus immediately upstream of the SNPA/G” and “All blastomeres (N=9) derived from control embryos were uniformly homozygous at the target site (*LDLRAP1* homo-WT) but heterozygous at the adjacent SNP locus (SNPA/G)” are misleading and confusing. These sentences should be changed to be accurate.

As indicated above, we revised the manuscript to avoid this confusion.

8. Line 155, “WT allele” should be clarified. Does it refer to “alleles without mutations caused by Cas9” or the allele with a G at the A/G SNP? The authors are suggested to distinguish the two regions by “cleavage site” (rather than target site, which in this

reviewer's opinion, is the region complementary to the sgRNA and include the A/G SNP locus), and "the A/G SNP".

We revised in the manuscript to make it clearer. We also clarified that WT is a maternal allele and paternal carries mutation.

9. *In Supplementary Fig.1C, the statement that the sgRNA targets "both wild type alleles" is inaccurate. In each embryo, only one wild type allele is present (in this case, "wildtype" means the G SNP).*

We revised in the Supplementary Figure.

10. *In Fig1C, 1D and related text, the authors are suggested to list both the types of mutation at the cleavage site and the A/G SNP, first maternal allele then paternal allele. For example, wt/Indel:G/A; Indel/Indel:A/A.*

We revised the Figure and related text, and added detail genotype information for each blastomeres in the Supplementary table 2.

Minor concerns:

1. *In Supplementary Fig.1A and 1C, the ssODN should be aligned to the reference sequence (with the sgRNA target site indicated) to clearly show the nucleotide changes.*

We aligned ssODN with the target region in the Figure as suggested.

2. *Please check typo: "Large deletions at the target locus ranging in size from 145bp to 3.8kb were found in in 14.3% of"*

Yes, we revised this sentence.

Reviewer #3 (Remarks to the Author):

In this study, Dr Lang and colleagues aim to characterize the spectrum of on-target mutations induced by gene editing of human embryos at the MII or zygote stage by molecular analysis at different developmental stages, cleavage stage (individual blastomeres collected on 3 of embryo development) and bulk DNA of derived ESCs.

To overcome the limitation of single cell analysis or pooled blastomeres from a single embryo that would prevent mosaicism assessment, the authors validated on-target edits seen in human embryos by analysis of embryonic stem cells (ESCs) derived from targeted embryos that provided ample DNA for detailed sequencing. Moreover, ECS analysis would not require whole-genome amplification, avoiding the well-known amplification biases.

Comparative analysis of blastomeres and ESC demonstrated a more frequent loss of heterozygosity

(LOH) in embryos than in ESCs suggesting false-positive readouts due to WGA. Some of these differences could be attributed to sampling differences but also to the artificial loss of one allele during faulty WGA.

For example, the frequency of homozygous indel genotypes in ESC clones was substantially lower than that found in blastomeres. However, this wasn't true for all loci investigated. For instance, the authors reported that the frequency of LOH at the heterozygous MYH7 locus, (homo-WT) in ESC was comparable to embryos (57.1% vs 67.4%).

Analysis of ESCs demonstrates that DSB repair in human preimplantation embryos produces an array of on-target modifications including indel mutations, large deletions, and gene conversions. The authors attempted to exclude aneuploidies by FISH and G-banding on ESC showing LOH at target loci, supporting that authentic LOH is likely caused by interallelic gene conversion in human embryos.

Fertilization, blastocyst development rates and ESC derivation are within the normal range for non-injected controls, suggesting that gene editing did not impair the preimplantation development of human embryos and subsequent ESC derivation.

This is a very interesting and well-developed paper from leading authors in the field. Overall, I think the data and discussion are good and reasonable, but several assays are missing, and the authors should reformulate their conclusions and discussion. Their conclusions are too strong for the limited number of techniques that they use, and more caution is needed in their interpretations. For instance, they make conclusions about ADO based on one SNP (mentioned in line 144).

We appreciate the reviewer's time and efforts in evaluating our manuscript and comments on the meaning of our work.

We provided additional data characterizing ADO in individual embryonic blastomeres (n=18) and single-cell fibroblasts (n=33) based on the analysis of 608 known heterozygous loci. We show that on average, 27% of amplified loci appeared as homozygous due to ADO in cloned embryos DNA. We hope that these new results more strongly support our conclusions.

These ADO results were part of a separate manuscript we are preparing to evaluate haploidy after SCNT (Alanis-Lobato *et al.*, 2021; Lee *et al.*, 2022) but we decided to pull these data from that manuscript and include them in the revised paper here.

They focus on the introduction of ADO due to WGA, but I think they should delve deeper into different types of ADO events, and at least mention that these may be different depending on the WGA technique for instance. Although it is undoubtedly true that WGA (particularly MDA-based WGA) results in frequent ADO, this is usually random across the genome and does not impact genotyping-based copy number analysis because ADO events are usually not contiguous along chromosomal regions; they are usually interspaced. This is a very well-known feature in clinical PGT, where ADO usually involve one marker of those

selected for linkage analysis. LOH of large and contiguous chromosomal regions is exceptionally rare following WGA.

The authors should comment on this. Also, was the karyotype of targeted embryos analyzed for aneuploidies? Did they check karyotypes of targeted blastomeres with PGTA?

I couldn't find this information.

We agree with the reviewer's suggestion and revised the text to indicate that ADO could vary depending on the WGA technique. We also added our reasoning behind using MDA-based WGA kit from Qiagen. We did not perform PGT-A for gene edited blastomeres. Unfortunately, pre-amplified single blastomere DNA (after MDA-based WGA) was not suitable for PGT-A assays.

The difference between LOH frequencies observed in blastomeres and ESC at the MYBPC3 locus and MYH7 locus doesn't seem to be statistically significant, given the small number of samples. Please quantify the statistical significance of this difference, for example providing a two-sample proportion test p-value (line 276,278)

We performed one-tailed Fisher's test as suggested and provided P values in the revised manuscript. For MYBPC3 locus, P values < 0.0001, while there has no significance in the MYH7 locus.

In general, the authors mention a lot of numbers, but they do not always explain clearly where these are coming from. For instance, line 118, they mention 15 embryos, but which group do these belong to?

Line 118, 15 embryos belong to the group that carried secondary gel bands (deletions). We revised the manuscript to make it clear.

Considering mosaicism in gene-edited human embryos, a minimum of 10 clonally propagated ESC lines were established from each embryo. BUT hESC line might have arisen from a few cells not representative of the entire embryonic constitution. The limitation of this approach is that ESCs represent a progeny of a few epiblast cells, indicating that the majority of genetic variants of mosaic embryos may not be preserved. Can the authors comment on this?

Yes, it is reasonable suggestion and ESCs do not "inherit" genetic variants of all cells from the embryo. We specified this limitation of ESC platform in the discussion.

Section "LOH in human embryos induced by DSBs at heterozygous loci"

- They utilize heterozygous sperm and therefore expect half of the embryos to originate from mutant sperm. For studying gene correction, it would be necessary to establish the true origin of the sperm for each embryo. It is always possible that a bias could be present (for instance, the mutant sperm may be more challenging to catch in a swim out).*

It is a possibility but we included a control fertilization group (no editing) and showed that in this particular case both mutant and WT sperm were equally “caught” for ICSI. Among 18 fertilized controls, 9 (50%) embryos were homozygous WT and 9 (50%) were heterozygous, carrying the wild-type maternal and mutant paternal alleles ($MYH7^{WT/Mut}$) (Supplementary Fig. 4b).

How did they track template used? Probably with the use of a synonymous SNP, but this information should be mentioned in the results.

Yes, all template ssODNs carried synonymous SNPs. We added this information as suggested.

- *Line 215: A much clearer explanation for these conclusions is necessary. It seems that only the presence of the homo-WT allele was used to detect LOH, which is insufficient. If more techniques were used these should be clearly stated. Another explanation for the homo-WT allele could be the preferential occurrence of a specific indel, for instance, due to microhomology-mediated end joining (MMEJ).*

These were mosaic embryos where at least one blastomere contained heterozygous $MYH7^{WT/Mut}$ genotype. We documented that some sister blastomeres from these mosaic embryos lost mutant allele and only WT allele was seen (homo-WT). In addition to sequencing approach, DNA from each these blastomeres were screened for large deletions by 3 independent long-range PCR. As indicated in the manuscript we reasoned that LOH in these samples could be due to gene conversion, unidentified deletions or artificial ADO caused by WGA.

Section "Validation of on-target modifications in ESCs derived from edited embryos"

- *Line 224: The rationale for targeting two sites simultaneously is lacking. To my understanding, for one target they attempt to correct a mutation, while for the other target they attempt to just introduce indels at a homozygous wild-type allele. This should be clarified. Preferentially, these different events should have been investigated in different embryos and not in the same embryo. An explanation for this is missing (in their discussion).*

We targeted simultaneously 2 loci in embryos that were used to derived ESCs purely for sake of reducing number of human oocytes/embryos needed to derive sufficient number of ESC lines. We added this point in the manuscript.

- *Line 231: I find the statement that they know that there was no negative selection too strong. The authors should make a comparison between the genetic events found in the stem cell lines and the embryos. If I understand correctly, only 57% (8/14) of the embryos displayed the WT allele which seems to suggest that theoretically only one embryo would be corrected. Again, the origin of the sperm (mutant or wild-type) is needed.*

In this sentence, we simply state that 48.3% ESC derivation efficiency in this study was exceptionally high compared to 30% average over many years in our laboratory. This suggests that gene edited embryos are equally capable of supporting ESC derivation and we did not find any negative selection.

- *Line 241-242: An explanation for what constitutes a small indel and what a large indel should be made.*

In this study, indels equal or larger than 100bp were designated as large, while smaller than 100bp as small. We added this designation in the revised manuscript.

- *Line 250: How do they know with certainty that no ADO is present? ADO can also be due to the presence of an SNP in the primer-binding site, so a total exclusion of ADO is not possible.*

We meant that ADO due to WGA can now be excluded since ESC DNA was not preamplified with WGA kit. We revised this section and included reviewer's suggestion that we cannot exclude ADO due to non-WGA amplification biases.

Section "LOH due to interallelic gene conversion"

- *Line 290: Why was this SNP assay not utilized for the embryo samples? The authors explain that they think that ADO can impede further LOH investigations, but this paper misses more proof for this.*

Most embryo samples were produced from different oocyte donors that did not carry informative SNPs. As indicated above we added additional results in the revised manuscript for embryos generated by SCNT from skin fibroblasts that carry 608 heterozygous loci. These results clearly demonstrate WGA introduces on average, 27% ADO.

Further comments

Line 375: To address the issue of large deletions, the authors utilized a FISH assay, however the resolution here is quite low (100-200 kb) therefore a lot of deletions, which they found with their long-range PCR assay in the embryos, would be missed. Therefore, a long-range PCR assay in stem cells would be more fitting. Why do they not utilize this technique for stem cells?

In addition to FISH and G-banding karyotyping, all ESCs and their subclones were screened for large deletions using the long-range PCR (up to 8-10Kb) similar to that described for embryos. No additional deletions were found except those reported in the manuscript. We clarified this in the revised manuscript.

In the stem cell studies they perform karyotyping, but this information is missing in the embryo part.

All embryos were disaggregated into single blastomeres and DNA from each individual blastomere was analyzed for on-target edits by sequencing after WGA. Therefore, karyotyping analysis of embryos was not feasible.

Figure 1D: Another graph type with more detail would be more fitting.

We revised Figure 1d as suggested.

Any difference in outcomes for immature, discarded and MII donated oocytes?

We only used mature MII oocytes donated by healthy egg donors in this study. Donated immature and discarded oocytes were used for preliminary experiments to test study protocols and to train personnel but these results were not included in the manuscript.

Fertilization rate, cleavage rate and embryo development metrics need to be reported in supplementary.

We appreciate reviewer's suggestion and added these metrics in the Supplementary table 7.

References:

- Alanis-Lobato, G., Zohren, J., McCarthy, A., Fogarty, N.M.E., Kubikova, N., Hardman, E., Greco, M., Wells, D., Turner, J.M.A., and Niakan, K.K. (2021). Frequent loss of heterozygosity in CRISPR-Cas9-edited early human embryos. *Proc Natl Acad Sci U S A* *118*. 10.1073/pnas.2004832117.
- Lee, Y., Trout, A., Marti-Gutierrez, N., Kang, S., Xie, P., Mikhalchenko, A., Kim, B., Choi, J., So, S., Han, J., et al. (2022). Haploidy in somatic cells is induced by mature oocytes in mice. *Commun Biol* *5*, 95. 10.1038/s42003-022-03040-5.
- Ma, H., Marti-Gutierrez, N., Park, S.W., Wu, J., Lee, Y., Suzuki, K., Koski, A., Ji, D., Hayama, T., Ahmed, R., et al. (2017). Correction of a pathogenic gene mutation in human embryos. *Nature* *548*, 413-419. 10.1038/nature23305.
- Papathanasiou, S., Markoulaki, S., Blaine, L.J., Leibowitz, M.L., Zhang, C.Z., Jaenisch, R., and Pellman, D. (2021). Whole chromosome loss and genomic instability in mouse embryos after CRISPR-Cas9 genome editing. *Nat Commun* *12*, 5855. 10.1038/s41467-021-26097-y.
- Zuccaro, M.V., Xu, J., Mitchell, C., Marin, D., Zimmerman, R., Rana, B., Weinstein, E., King, R.T., Palmerola, K.L., Smith, M.E., et al. (2020). Allele-Specific Chromosome Removal after Cas9 Cleavage in Human Embryos. *Cell* *183*, 1650-1664 e1615. 10.1016/j.cell.2020.10.025.

Reviewers' Comments:

Reviewer #1:

Remarks to the Author:

The Authors have satisfactorily addressed all my comments and considerably improved the manuscript, by performing additional experiments and improving the clarity of text and figures. Therefore, I am now supportive of publication of this manuscript in Nature Communications.

Reviewer #2:

Remarks to the Author:

The authors have adequately addressed my concerns.

Reviewer #3:

Remarks to the Author:

The authors have performed a comprehensive review and answered properly to all my questions. No further comments.